# SELF-SUPERVISED LOGIT ADJUSTMENT

## ABSTRACT

Self-supervised learning (SSL) has achieved tremendous success on various well curated datasets in computer vision and natural language processing. Nevertheless, it is hard for existing works to capture transferable and robust features, when facing the long-tailed distribution in the real-world scenarios. The attribution is that plain SSL methods to pursue *sample-level uniformity* easily leads to the distorted embedding space, where head classes with the huge sample number dominate the feature regime and tail classes passively collapse. To tackle this problem, we propose a novel Self-Supervised Logit Adjustment ($S^2LA$) method to achieve the *category-level uniformity* from a geometric perspective. Specially, we measure the geometric statistics of the embedding space to construct the calibration, and jointly learn a surrogate label allocation to constrain the space expansion of head classes and avoid the passive collapse of tail classes. Our proposal does not alter the setting of SSL and can be easily integrated into existing works in a low-cost manner. Extensive results on a range of benchmark datasets show the effectiveness of $S^2LA$ with high tolerance to the distribution skewness.

## 1 INTRODUCTION

Recent years have witnessed a great success of self-supervised learning (Doersch et al., 2015; Wang & Gupta, 2015; Chen et al., 2020; Caron et al., 2020). The rapid advances behind this paradigm benefit from the elegant training on data without annotations, which can be acquired in a large-volume and low-cost way. However, the real-world natural sources usually exhibit the long-tailed distribution (Reed, 2001), and directly applying existing self-supervised learning methods will lead to the distorted embedding space, where the majority dominates the feature regime (Zhang et al., 2021) and the minority collapses (Mixon et al., 2022). With the increasing attention on machine learning fairness in the recent years, it becomes a trend to explore self-supervised long-tailed learning (Yang & Xu, 2020; Liu et al., 2021; Jiang et al., 2021; Zhou et al., 2022).

Compared with the flourishing supervised long-tailed learning (Kang et al., 2019; Yang & Xu, 2020; Menon et al., 2021), the self-supervised counterpart is underexplored as an emerging direction. Existing explorations for self-supervised learning in long-tailed context are from three perspectives: *data perspective*, *model perspective* and *loss perspective*. In the data perspective, BCL (Zhou et al., 2022) leverages the memorization effect of deep neural networks (DNNs) to drive an instance-wise augmentation, which learns a better trade-off between head classes and tail classes in representation learning. In the model perspective, SDCLR (Jiang et al., 2021) contrasts the feature encoder and its pruned counterpart to discover hard examples that mostly covers the samples from tail classes, and efficiently enhance the learning preference towards tail classes. In the loss perspective, the reweighting mechanism like rwSAM (Liu et al., 2021) that adopts a data-dependent sharpness-aware minimization scheme, can be applied to explicitly regularize the loss surface. However, in terms of the current performance to self-supervised long-tailed learning, the potential of the loss perspective has not been sufficiently set off, with which in comparison in supervised long-tailed learning, logit adjustment (Menon et al., 2021) of the same perspective has conquered a range of methods.

We dive into the loss perspective and explore to understand *"Why the conventional contrastive learning underperforms in self-supervised long-tailed learning?"* To answer this question, let us consider two types of representation uniformity: (1) *Sample-level uniformity*. As proof in (Wang & Isola, 2020), contrastive learning targets to distribute the representation of data points uniformly in the embedding space. Then, the feature span of each category is proportional to their corresponding sample number. (2) *Category-level uniformity*. This uniformity pursues to split the region equally

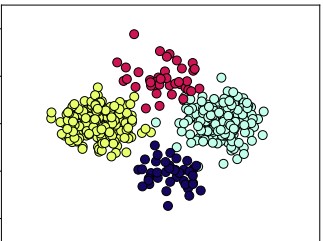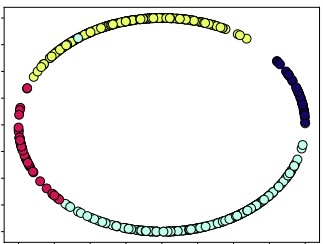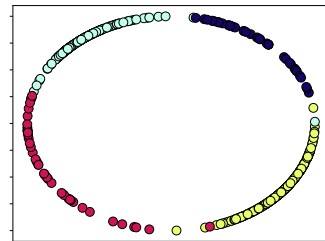

Figure 1: Comparison of S²LA and the plain SSL method on a 2-D imbalanced synthetic dataset. (Left) Visualization of the 2-D synthetic dataset. (Middle) The embedding distribution of each category learnt by the plain SSL method is approximately proportional to the sample number. (Right) S²LA reduces the adverse effect of class imbalance and approaches to the category-level uniformity.

for different categories without considering their corresponding sample number (Papyan et al., 2020; Graf et al., 2021). In the case of the class-balanced scenarios, the former uniformity naturally implies the latter uniformity and induces the equivalent separability for classification. However, in the class-imbalanced cases, especially in long-tailed setting, sample-level uniformity leads to the undesired feature regime of head classes due to its dominant sample proportion. In comparison, category-level uniformity constrains the greedy expansion of head classes and prevents the passive collapse of tail classes, which is more benign to the downstream classification (Graf et al., 2021; Fang et al., 2021; Li et al., 2022). Unfortunately, there is no support regarding category-level uniformity in contrastive learning losses, which provides an answer to the question arisen at the beginning.

Inspired by logit adjustment (LA) for supervised long-tailed learning, we propose a novel method, termed as *Self-Supervised Logit Adjustment* (S²LA), to calibrate self-supervised long-tailed learning from the geometric perspective. Specially, unlike LA that requires the class distribution available, S²LA uses a constant Simplex ETF to measure the geometric characteristics of the embedding space for adjustment. Together with a surrogate label allocation to compute the target, we can then explicitly compress the greedy space expansion of head classes and avoid the passive collapse of tail classes. Their alternation refers to an ordinary balancing and an efficient optimal-transport problem, which dynamically approaches towards the category-level uniformity. In Figure 1, we give a toy experiment to compare the learnt representation without and with S²LA in the embedding space.

The contribution can be summarized as follows,

1. We are among the first attempts to study the drawback of the contrastive learning loss in self-supervised long-tailed context and point out that the resulting sample-level uniformity is an intrinsic limitation, driving our exploration for category-level uniformity (Section 4).

2. We develop a novel *Self-Supervised Logit Adjustment* (Figure 2), which dynamically adjusts the embedding distribution to calibrate the geometric statistics and conducts a surrogate label allocation for category-level uniformity in an efficient end-to-end manner.

3. Our method can be easily plugged into previous methods of self-supervised long-tailed learning (Eq. 7). Extensive experiments on a range of benchmark datasets demonstrate the consistent improvement of our S²LA with high tolerance to the distribution skewness.

## 2 RELATED WORKS

**Self-Supervised Long-tailed Learning.** There are several recent explorations devoted to this directions from data, model and loss perspectives. BCL (Zhou et al., 2022) leverages the memorization effect of DNNs to drive an instance-wise augmentation, which enhances the learning of tail samples. SDCLR (Jiang et al., 2021) constructs a self-contrast between model and its pruned counterpart to learn more balanced representation. Classic Focal loss (Lin et al., 2017) leverages the loss statistics to putting more emphasis on the hard examples, which has been applied to self-supervised long-tailed learning (Zhou et al., 2022). SeLa (Asano et al., 2020) is the first attempt to cast the unsupervised clustering as an optimal transport problem and leverage a uniform prior on the class-imbalance data. rwSAM (Liu et al., 2021) proposes to penalize loss sharpness in a reweighting manner to similarly calibrate class-imbalance learning. However, the potential of the loss perspective has not been set-off due to the intrinsic limitation of the conventional contrastive learning loss.

**Hyperspherical Uniformity.** The distribution uniformity has been extensively explored from the physic area, *e.g.*, Thomson problem (Thomson, 1904; Smale, 1998), to machine learning area like some kernel-based extensions *e.g.*, Riesz s-potential (Hardin & Saff, 2005; Liu et al., 2018) or Gaussian potential (Cohn & Kumar, 2007; Borodachov et al., 2019; Wang & Isola, 2020). Some recent explorations regarding features of DNNs (Papyan et al., 2020; Fang et al., 2021; Mixon et al., 2022) discover a terminal training stage when the embedding collapses to the geometric means of the classifier *w.r.t.* each category. Specially, these optimal class means specify a maximum separation structure, termed as Simplex Equiangular Tight Frame, and achieve the perfect uniform distribution under some dimensional constraints (Zhu et al., 2021; Yang et al., 2022; Kasarla et al., 2022). In this paper, we incorporate Simplex ETF into contrastive learning and leverage the specific structure as a uniform prior to capture the geometric statistics for calibration of the embedding distribution.

## 3 PRELIMINARIES

### 3.1 PROBLEM FORMULATION

Given a dataset $\mathcal{D}$, for each $(\boldsymbol{x}, \boldsymbol{y}) \in \mathcal{D}$, the input $\boldsymbol{x} \in \mathbb{R}^m$ and the associated label $\boldsymbol{y} \in \{1, \ldots, L\}$. We define the imbalanced ratio as $\max_i p(\boldsymbol{y} = i)/\min_i p(\boldsymbol{y} = i)$. In SSL, the ground-truth $\boldsymbol{y}$ is not accessed and the goal is to transform an image to an embedding via DNNs $f_\theta : \mathbb{R}^m \to \mathbb{R}^d$. In the linear probing evaluation, we construct a supervised learning task with balanced datasets. A linear classifier $g$ is built on top of the feature extractor $f_\theta$ to produce prediction, namely, $g(f_\theta(\boldsymbol{x}))$.

### 3.2 LOGIT ADJUSTMENT IN SUPERVISED LONG-TAILED LEARNING

Logit adjustment manipulates the logits of prediction to remove the adverse effect of class-imbalance, which has shown efficiency in various supervised long-tailed explorations (Provost, 2000; Brodersen et al., 2010; Ren et al., 2020; Menon et al., 2021). Let $p(\boldsymbol{y}), p_{bal}(\boldsymbol{y})$ be the label distribution of the long-tailed training dataset and the balanced test set, we can derive $p(\boldsymbol{y}|\boldsymbol{x}) \propto p(\boldsymbol{x}|\boldsymbol{y})p(\boldsymbol{y})$ and $p_{bal}(\boldsymbol{y}|\boldsymbol{x}) \propto p(\boldsymbol{x}|\boldsymbol{y})p_{bal}(\boldsymbol{y})$ based on the Bayes' theorem. Assuming class-conditional probabilities $p(\boldsymbol{x}|\boldsymbol{y})$ are the same on training and test set, we have $p(\boldsymbol{y}|\boldsymbol{x}) \propto p_{bal}(\boldsymbol{y}|\boldsymbol{x})p(\boldsymbol{y})$. This indicates that learning under long-tailed data leads to a prediction shift proportional to $p(\boldsymbol{y})$ compared with the optimal classifier on the uniform test set. To this end, logit adjustment adds an offset on the logits to calibrate the statistical shift, namely,

$$p_{bal}(\boldsymbol{y}|\boldsymbol{x}) \propto p(\boldsymbol{y}|\boldsymbol{x})/p(\boldsymbol{y}) \propto \text{softmax}(s(\boldsymbol{x}) - \log p(\boldsymbol{y})), \tag{1}$$

where $s(\boldsymbol{x})$ denotes the logits of the training samples and $\text{softmax}(\cdot)$ is the Softmax function. Logit adjustment are well grounded to be Fisher consistent for minimising the balanced error (Menon et al., 2021), thus extended to a range of class-imbalanced scenarios (Ren et al., 2022).

### 3.3 SIMPLEX EQUIANGULAR TIGHT FRAME

Neural collapse (Mixon et al., 2022) describes a phenomenon that with the training, the geometric centroid of representation progressively collapses to the optimal classifier parameter *w.r.t.* each category. The collection of these points builds a special geometric structure, termed as Simplex Equiangular Tight Frame (ETF). Some study that shares the similar spirit is also explored regarding the maximum separation structure (Kasarla et al., 2022). We present its formal definition as follows.

**Definition 3.1.** A Simplex ETF is a collection of points in $\mathbb{R}^d$ specified by the columns of the matrix:

$$\mathbf{M} = \sqrt{\frac{K}{K-1}} \mathbf{U}(\mathbf{I}_K - \frac{1}{K}\mathbb{1}_K \mathbb{1}_K^{\mathrm{T}}) \tag{2}$$

where $\mathbf{I}_K \in \mathbb{R}^{K \times K}$ is the identity matrix and $\mathbb{1}_K$ is the $K$-dimensional ones vector. $\mathbf{U} \in \mathbb{R}^{d \times K}$ is the patial orthogonal matrix such that $\mathbf{U}^\top \mathbf{U} = \mathbf{I}_K$ and it satisfys $d \geq K$. All vectors in a Simplex ETF have the same pair-wise angle, *i.e.*, $\boldsymbol{m}_i \boldsymbol{m}_j = -\frac{1}{K-1}, i \neq j$. The pioneering work (Yang et al., 2022) shows Simplex ETF as a linear classifier combined with neural networks is robust to class-imbalanced learning in the supervised setting. On the opposite, our motivation is to make self-supervised learning robust to the class-imbalance data, which requires the pursuit in the embedding

space intrinsically switching from the sample-level uniformity to the category-level uniformity. The Simplex ETF is a tool to measure the gap between the category-level uniformity and the sample-level uniformity, which is then transformed as the supervision feedback to the training.

# 4 SELF-SUPERVISED LOGIT ADJUSTMENT

Plain SSL methods encourage the sample-level uniformity in the embedding space, yielding the undesired property in long-tailed context. Specially, head classes will dominate the feature regime along with the collapse of tail classes. To solve this problem, one straightforward idea is to counteract the long-tailed effect like logit adjustment in supervised long-tailed learning. As shown in Figure 1, we observe that the geometric calibration will benefit the feature balancedness. However, the core challenge in self-supervised learning is that there is no class-distribution prior available to characterize the imbalance degree. To address this issue, we propose to use Simplex ETF to measure the embedding space, and then the captured geometric statistics are used to properly adjust the embedding distribution alongside a surrogate label allocation. Concretely, we define an constant classifier $\mathbf{M} \in \mathbb{R}^{d \times K}$ following Eq. 2, and then compute the geometric label of each sample:

$$\boldsymbol{t} = \arg\max_i h_i(\boldsymbol{x}), \quad \text{where} \ \ h_i(\boldsymbol{x}) = \mathbf{M}_i^\top f_\theta(\boldsymbol{x}).$$

On the basis of such a positional indicator, we can further derive a population-level statistic to characterize the category-level uniformity, i.e., $p(\boldsymbol{t} = i) = \mathbb{E}_{\boldsymbol{x} \sim \mathcal{D}}[\mathbb{1}(\boldsymbol{t} = i)], i = 1, \ldots, K$, where $\mathbb{1}(\cdot)$ denotes the indicator function with the value 1 when its argument is true and 0 otherwise. In the balanced setting, we maintain an approximately uniform distribution while in long-tailed context, we face a more skewed geometric distribution for the category-level uniformity.

**Logit Adjustment.** Different from the ordinary logit adjustment that removes the class imbalance in $p(\boldsymbol{t})$ and exacerbates the space expansion, here we aim to shrink the feature span of head classes for the category-level uniformity, and thus reversely adjust the logits, namely:

$$\tilde{\boldsymbol{t}} = \arg\max p(\boldsymbol{t}|\boldsymbol{x})p(\boldsymbol{t})^\tau = \arg\max h(\boldsymbol{x}) + \tau \cdot \log p(\boldsymbol{t}) \tag{3}$$

where $p(\boldsymbol{t}|\boldsymbol{x}) = \text{softmax}(h(\boldsymbol{x}))$ denotes the output probabilities of the geometric classifier and $\tau$ is a hyper-parameter to temper the adjustment. We can derive $p\left(\tilde{\boldsymbol{t}}|\boldsymbol{x}\right) \propto p(\boldsymbol{x}|\boldsymbol{t})p(\boldsymbol{t})^{1+\tau}$ based on Bayes' theorem, which refers to an exponential growth over the original distribution $p(\boldsymbol{t}|\boldsymbol{x}) \propto p(\boldsymbol{x}|\boldsymbol{t})p(\boldsymbol{t})$. This fulfills an elastic adjustment on the embedding skewness. Logit adjustment is to strengthen the estimation of samples from head classes so that they can be discovered as many as possible, which is then used to suppress the feature regime of head samples for the category-level uniformity. This is different from logit adjustment in supervised long-tailed learning, which is to calibrate the quantity bias towards different classes of samples in the logit space.

However, since there is no ground-truth annotation of each sample, straightforward instance-level adjustment will lead to performance degradation with unreliable predictions, as empirically evidenced in Table 10. To handle these problems, we learn a surrogate label allocation together with logit adjustment from the population level. Concretely, we first estimate the adjusted statistic $p\left(\tilde{\boldsymbol{t}}\right)$ based on Eq. 3, i.e., $p\left(\tilde{\boldsymbol{t}} = i\right) = \mathbb{E}_{\boldsymbol{x} \sim \mathcal{D}}[\mathbb{1}\left(\tilde{\boldsymbol{t}} = i\right)], i = 1, \ldots, K$, and then propose the following objective to generate label posterior $q(\boldsymbol{t}|\boldsymbol{x})$ with the desired marginal distribution $p(\tilde{\boldsymbol{t}})$:

$$\min_{p,q} -\frac{1}{|\mathcal{D}|} \sum_{x \sim \mathcal{D}} q(\boldsymbol{t}|\boldsymbol{x}) \log p(\boldsymbol{t}|\boldsymbol{x}), \quad \text{s.t.} \ \ \mathbb{E}_{\boldsymbol{x} \sim \mathcal{D}}[q(\boldsymbol{t}|\boldsymbol{x})] = p\left(\tilde{\boldsymbol{t}}\right) \tag{4}$$

where the constraint guarantees the adjusted population-level statistics. Naive jointly optimizing Eq. 4 easily falls into a trivial solution and causes severe performance degeneration, as shown in Table 10. Considering this, we resort to solving an optimal transport problem for label allocation on $q(\boldsymbol{t}|\boldsymbol{x})$ and then optimize the model parameters.

**Label Allocation.** We first reformulate Eq. 4 as an optimal transport (OT) problem. Denote the assignments $Q \in \mathbb{R}^{K \times N}$ for distributing $N$ samples to $K$ classes with the minimal transportation cost and let $C = -\log P$ to retain the equivalent objective, where $P \in \mathbb{R}^{K \times N}$ are the joint probabilities derived from $q(\boldsymbol{t}|\boldsymbol{x})$. We can rewrite Eq. 4 as the following form:

$$\min_{Q \in \mathbb{R}^{K \times N}} \langle Q, C \rangle, \quad \text{s.t.} \ \ Q \cdot \mathbb{1}_N = p\left(\tilde{\boldsymbol{t}}\right), \ Q^\mathrm{T} \cdot \mathbb{1}_K = \mathbb{1}_N \tag{5}$$

Figure 2: Illustration of S²LA on the top of SSL. We utilize the Simplex ETF structure to measure the embedding space, and the captured geometric statistics are used for the target calibration in logit adjustment. Simultaneously, an optimal transport optimization for label allocation is performed together with adjustment to avoid the trivial solutions. The additional model parameters introduced by S²LA are trained with SSL in an efficient manner.

This objective can be efficiently solved by adopting *Sinkhorn-Knopp algorithm* (Cuturi, 2013) with an additional entropy regularization to the objective, *i.e.*, $\min_Q \langle Q, C \rangle - \frac{1}{\lambda} \mathrm{H}(Q)$. Here, we define the entropy of $Q$ as $\mathrm{H}(Q) = -\sum_i \sum_j Q_{ij} \log Q_{ij}$. We can then obtain the solution as:

$$Q = \mathrm{diag}(\boldsymbol{u}) \exp\left(-\lambda C\right) \mathrm{diag}(\boldsymbol{v}) \tag{6}$$

where $\boldsymbol{u}, \boldsymbol{v}$ are two non-negative vectors of $\mathbb{R}^d$. We iteratively update $\boldsymbol{u}$ and $\boldsymbol{v}$ to reach the desired row and column marginals of $Q$. The coefficient $\lambda$ controls the smoothness of the low-cost assignments. As $\lambda \to \infty$, we can approach the optimum of the OT problem at the cost of convergence speed. We keep $\lambda$ at a relative high level to maintain good clustering results. To explain the merits of the population-level label allocation, we compute the NMI score between $\tilde{t}$ or $q(t|x)$ and the oracle label in Table 11. Accorrding to the comparison of their NMIs, we can understand the roughly adjusted $\tilde{t}$ can be noisy and shows the weak correlation with the oracle label, while $q(t|x)$ reallocated by optimal transport based on the population statistic of $\tilde{t}$ effectively improves the correlation.

In practice, we estimate $p(\boldsymbol{t})$ and $p(\tilde{\boldsymbol{t}})$ over the training dataset and learn the label allocation $q(\boldsymbol{t}|\boldsymbol{x})$ in the mini-batch manner. Specifically, we compute the class prior $p(\boldsymbol{t})$ and the adjusted $p(\tilde{\boldsymbol{t}})$ at the beginning of every epoch as the population-level statistic will not change much in a few mini-batches. Besides, we maintain a momentum update mechanism to track the output logits of each sample to stabilize the training, *i.e.*, $h^m(\boldsymbol{x}) \leftarrow \beta h^m(\boldsymbol{x}) + (1-\beta)h(\boldsymbol{x})$. Given the surrogate labels $q(\boldsymbol{t}|\boldsymbol{x})$, we can directly optimize $p(\boldsymbol{t}|\boldsymbol{x})$ by minimising the cross-entropy loss, which can be considered as the alignment to the desired geometric distribution. We illustrate S²LA in Figure 2 and summarize the complete procedure in Algorithm 1 in the appendix.

**Overall Objective.** S²LA is compatible with the general self-supervised learning methods as it is constructed on the surrogate geometric analysis to rectify the distorted embedding space in long-tailed context. For example, if we build S²LA on SimCLR, the overall objective is defined as:

$$\mathcal{L} = \frac{1}{|\mathcal{B}|} \sum_{b \in \mathcal{B}} \frac{1}{|\mathcal{X}_b|} \sum_{\boldsymbol{x} \in \mathcal{X}_b} \left( -\log \frac{\exp\left(f_\theta(\boldsymbol{x})^\top f_\theta(\boldsymbol{x}^+)/\gamma\right)}{\sum_{\boldsymbol{x}^- \in \mathcal{X}_b^- \cup \{\boldsymbol{x}^+\}} \exp\left(f_\theta(\boldsymbol{x})^\top f_\theta(\boldsymbol{x}^-)/\gamma\right)} - q(\boldsymbol{t}|\boldsymbol{x}^+) \log p(\boldsymbol{t}|\boldsymbol{x}) \right) \tag{7}$$

where $\boldsymbol{x}, \boldsymbol{x}^+$ is set as different augmented view of the same input image and $\boldsymbol{x}^-$ denotes the rest images in the mini-batch. $\mathcal{X}_b$ is the sampled batch and $\mathcal{X}_b^-$ is the negative sample set with $b$ as the batch index. $\gamma$ represents the temperature hyper-parameter.

*Remark* 4.1. Although S²LA iteratively infers the surrogate labels and applies them for re-balanced training, it incurs only a small amount of computational or memory overhead. Specifically, the standard optimization of deep neural networks requires forward and backward step in each mini-batch update with the time complexity as $\mathcal{O}(M\Lambda)$, where $M$ is the mini-batch size and $\Lambda$ is the parameter size. At the parameter level, we add an additional classifier with the complexity as $\mathcal{O}(MKd)$, where $K$ is the class number and $d$ is the embedding dimension. Besides, Sinkhorn-Knopp algorithm only refers to a simple matrix-vector multiplication $u \leftarrow r./\exp(-\lambda C)v, v \leftarrow c./[\exp(-\lambda C)]^\top u$, whose complexity is $\mathcal{O}(L(M + K + MK))$ with the iteration step $L$. The complexity incurred in the momentum update is $\mathcal{O}(MK)$. Since $K, d$ and $L$ is significantly smaller than the model parameter $\Lambda$ of a million scale, the computational overhead involved in S²LA is negligible compared to $\mathcal{O}(M\Lambda)$. The additional storage is the vector $h^m(\boldsymbol{x}) \in \mathbb{R}^{K \times M}$, which is also negligible to the

memory usage. In total, our method incurs only a small computation or memory overhead and can thus be plugged into baseline methods in a low-cost manner. Moreover, we empirically compare the computational cost in Table 6 and observe that our method only incurs a relatively lightweight overhead compared with the baseline methods on a range of datasets.

## 4.1 DISCUSSION

**Relation to Supervised Logit Adjustment.** In supervised long-tailed learning, the optimal class-probabilities $p_{bal}(\boldsymbol{y}|\boldsymbol{x})$ follow the uniform distribution, which is aligned with the underlying ground-truth distribution $p_{bal}(\boldsymbol{y})$ on the test set. In contrast, the surrogate label distribution $p(\boldsymbol{t})$ in SSL encodes the positional information, which is better to be imbalanced. In our algorithm, we push away $p(\boldsymbol{t}|\boldsymbol{x})$ from $p_{bal}(\boldsymbol{t}|\boldsymbol{x})$ by replacing the division operation of Eq. 1 with multiplication, *i.e.*, $p_{imb}(\boldsymbol{t}|\boldsymbol{x}) \propto p(\boldsymbol{t}|\boldsymbol{x})p(\boldsymbol{t})^{\tau}$. Although it seems that we reversely enlarges the skewness of the geometric prediction, the surrogate supervised learning via Eq. 5 actually guarantees the space shrinking of head classes and the space expansion of tail classes.

**Difference from Unsupervised Clustering.** Compared with previous explorations (Asano et al., 2020; Caron et al., 2020), the uniqueness of S$^2$LA lies in the following two aspects: (1) *Geometric Classifier*. The pioneering works mainly resort to a learnable classifier to perform clustering, which can easily be distorted in the long-tailed scenarios (Fang et al., 2021). Built on the maximum separation classifier, our method is capable to provide clustering results with clear geometric interpretations. (2) *Adjustable Class Prior*. The class prior are assumed to be uniform among the previous attempts. When moving to the long-tailed case, this assumption will cause the undesired sample-level uniformity. In contrast, our methods can potentially cope with any distribution with a proper adjustment. In Section 5.4, we empirically demonstrate the priority of our S$^2$LA.

## 5 EXPERIMENTS

### 5.1 EXPERIMENTAL SETUP

**Baselines.** We choose four representative state-of-the-art methods: one is a plain SSL method SimCLR (Chen et al., 2020) and the other three are Focal (Lin et al., 2017), SDCLR (Jiang et al., 2021) and BCL (Zhou et al., 2022) respectively from *loss*, *model* and *data* perspectives.

**Implementation Details.** Following previous works (Jiang et al., 2021; Zhou et al., 2022), we use ResNet-18 as the backbone for small-scale dataset (CIFAR-100-LT) and ResNet-50 for large-scale datasets (ImageNet-LT/Places-LT). For experiments on CIFAR-100-LT, we train model with the SGD optimizer for 1000 epochs with the batch size 512, momentum 0.9 and weight decay factor $5 \times 10^{-4}$ and use the cosine annealing decay with the initial learning rate as 0.5. For experiments on ImageNet-LT and Places-LT, we only train for 500 epochs with the batch size 256 and decrease the weight decay factor to $1 \times 10^{-4}$. Other pre-training setups like the data augmentation and projector structure follow (Jiang et al., 2021; Zhou et al., 2022). For $\tau$ in S$^2$LA, we choose from [0.03,0.04,0.05,0.06,0.07] on CIFAR-100-LT and 0.02 for Places-LT and ImageNet-LT. We set the dimension $K$ as 100 and the Sinkhorn iteration as 300. As S$^2$LA is combined with baselines, warm-up of 500 epochs on CIFAR-100-LT and 400 epochs on ImageNet-LT and Places-LT is applied.

**Evaluation Metrics.** Following (Jiang et al., 2021; Zhou et al., 2022), *linear probing* on a balanced dataset is used for evaluation. We conduct full-shot evaluation on CIFAR-100-LT and few-shot evaluation on ImageNet-LT and Places-LT. We report the model performance and the standard deviation among three disjoint groups, *i.e.*, many/medium/few partitions (Liu et al., 2019).

### 5.2 SELF-SUPERVISED LONG-TAILED LEARNING PERFORMANCE

**CIFAR-100-LT.** In Table 1, we compare methods with and without S$^2$LA, and analyze as follows.

(1) *Overall performance.* S$^2$LA shows a consistent gain in many/medium/few groups, yielding a overall performance improvements averaging as 2.32%, 2.33% and 1.57% under different imbalance ratios. In particular, compared with previous state-of-the-art BCL, with S$^2$LA, BCL further achieves improvements by 1.20%, 1.82% and 1.22% on CIFAR-100-LT-R100/R50/R10.

Table 1: Linear probing on CIFAR-100-LT with different imbalanced ratios (100,50,10). Std means the group-level performance standard deviation and Avg is the average accuracy of full test set. Std represents a balancedness measure to quatify the variance among three specified groups. Here, Many/Med/Few indicate the fine-grained groups according to the class cardinality.

| Method | CIFAR-100-LT-R100 | | | | | CIFAR-100-LT-R50 | | | | | CIFAR-100-LT-R10 | | | | |
|---|---|---|---|---|---|---|---|---|---|---|---|---|---|---|---|
| | Many | Med | Few | Std | Avg | Many | Med | Few | Std | Avg | Many | Med | Few | Std | Avg |
| SimCLR | 54.97 | 49.39 | 47.67 | 3.82 | 50.72 | 56.00 | 50.48 | 50.12 | 3.30 | 52.24 | 57.85 | 55.06 | 54.03 | 1.98 | 55.67 |
| +S$^2$LA | 57.38 | 52.27 | 52.12 | 2.99 | 53.96 | 58.88 | 53.00 | 54.27 | 3.09 | 55.42 | 59.26 | 56.91 | 55.85 | 1.75 | 57.36 |
| **Improv.** | **+2.41** | **+2.88** | **+4.45** | **-0.82** | **+3.24** | **+2.88** | **+2.52** | **+4.15** | **-0.20** | **+3.18** | **+1.41** | **+1.85** | **+1.82** | **-0.23** | **+1.69** |
| Focal | 54.24 | 49.58 | 49.21 | 2.80 | 51.04 | 55.40 | 51.14 | 50.02 | 2.84 | 52.22 | 58.18 | 55.82 | 54.64 | 1.80 | 56.23 |
| +S$^2$LA | 57.01 | 52.93 | 51.74 | 2.76 | 53.92 | 57.97 | 53.55 | 53.58 | 2.54 | 55.06 | 60.06 | 56.79 | 57.24 | 1.77 | 58.05 |
| **Improv.** | **+2.77** | **+3.35** | **+2.53** | **-0.04** | **+2.88** | **+2.57** | **+2.41** | **+3.56** | **-0.30** | **+2.84** | **+1.88** | **+0.97** | **+2.60** | **-0.03** | **+1.82** |
| SDCLR | 57.32 | 50.70 | 50.45 | 3.90 | 52.87 | 57.50 | 51.85 | 52.15 | 3.18 | 53.87 | 58.47 | 54.79 | 52.97 | 2.80 | 55.44 |
| +S$^2$LA | 57.44 | 52.85 | 54.06 | 2.38 | 54.81 | 58.47 | 53.88 | 53.58 | 2.74 | 55.34 | 59.21 | 56.06 | 55.58 | 1.97 | 56.97 |
| **Improv.** | **+0.12** | **+2.15** | **+3.61** | **-1.52** | **+1.94** | **+0.97** | **+2.03** | **+1.43** | **-0.44** | **+1.47** | **+0.74** | **+1.27** | **+2.61** | **-0.83** | **+1.53** |
| BCL | 59.15 | 54.82 | 55.30 | 2.37 | 56.45 | 59.44 | 54.73 | 57.30 | 2.36 | 57.18 | 60.41 | 57.15 | 59.76 | 1.73 | 59.12 |
| +S$^2$LA | 59.50 | 55.73 | 57.67 | 1.89 | 57.65 | 60.82 | 57.58 | 58.55 | 1.66 | 59.00 | 61.41 | 59.27 | 60.30 | 1.07 | 60.34 |
| **Improv.** | **+0.35** | **+0.91** | **+2.37** | **-0.49** | **+1.20** | **+1.38** | **+2.85** | **+1.25** | **-0.70** | **+1.82** | **+1.00** | **+2.12** | **+0.54** | **-0.66** | **+1.22** |

Table 2: Linear probing on ImageNet-LT and Places-LT. Similarly, Std means the group-level performance standard deviation and Avg is the average accuracy of full test set. Std represents a balancedness measure to quatify the variance among three specified groups. Here, Many/Med/Few indicate the fine-grained partitions according to the class cardinality.

| Method | ImageNet-LT | | | | | Places-LT | | | | |
|---|---|---|---|---|---|---|---|---|---|---|
| | Many | Med | Few | Std | Avg | Many | Med | Few | Std | Avg |
| SimCLR | 41.69 | 33.96 | 31.82 | 5.19 | 36.65 | 31.98 | 34.05 | 35.63 | 1.83 | 33.61 |
| +S$^2$LA | 41.53 | 36.35 | 35.84 | 3.15 | 38.28 | 32.46 | 35.03 | 36.14 | 1.89 | 34.33 |
| **Improv.** | **-0.16** | **+2.39** | **+4.02** | **-2.04** | **+1.63** | **+0.48** | **+0.98** | **+0.51** | **+0.06** | **+0.72** |
| Focal | 42.04 | 35.02 | 33.32 | 4.62 | 37.49 | 31.69 | 34.33 | 35.73 | 2.05 | 33.65 |
| +S$^2$LA | 42.55 | 36.75 | 36.28 | 3.49 | 38.92 | 32.40 | 35.14 | 36.49 | 2.08 | 34.42 |
| **Improv.** | **+0.51** | **+1.73** | **+2.96** | **-1.13** | **+1.43** | **+0.71** | **+0.81** | **+0.76** | **+0.03** | **+0.77** |
| SDCLR | 40.87 | 33.71 | 32.07 | 4.68 | 36.25 | 32.17 | 34.71 | 35.69 | 1.82 | 33.99 |
| +S$^2$LA | 41.92 | 36.53 | 36.04 | 3.26 | 38.53 | 32.78 | 35.60 | 36.18 | 1.82 | 34.70 |
| **Improv.** | **+1.05** | **+2.82** | **+3.97** | **-1.42** | **+2.28** | **+0.61** | **+0.89** | **+0.49** | **0.00** | **+0.71** |
| BCL | 42.92 | 35.89 | 33.93 | 4.73 | 38.33 | 32.69 | 35.37 | 37.18 | 2.26 | 34.76 |
| +S$^2$LA | 43.22 | 38.16 | 36.96 | 3.32 | 39.95 | 33.22 | 36.00 | 37.62 | 2.23 | 35.32 |
| **Improv.** | **+0.30** | **+2.27** | **+3.03** | **-1.40** | **+1.62** | **+0.53** | **+0.63** | **+0.44** | **-0.03** | **+0.56** |

(2) *Representation balancedness.* Previously, we claim S$^2$LA help compress the expansion of head classes and avoid the passive collapse of tail classes, yielding a more balanced representation distribution. To certify this aspect, we compute the difference in performance among many/medium/few groups, namely, their groupwise standard deviation. Our method provides 1.41%/2.32%/3.24%, 1.95%/2.45%/2.60% and 1.26%/1.55%/1.89% improvements *w.r.t.* many/medium/few group on CIFAR-100-LT-R100/R50/R10 with more preference to the tail classes. Therefore, S$^2$LA substantially improves the standard deviation by 0.72/0.41/0.44 on CIFAR-100-LT-R100/R50/R10.

(3) *Regarding skewness.* Our method can be generalized to practical distributions with varying levels of skewness as the algorithm does not refer to any information about the distribution prior.

**ImageNet-LT and Places-LT.** Table 2 shows the comparison of different methods on large-scale dataset ImageNet-LT and Places-LT, in which we have similar observations. As can be seen, on more challenging real-world data, S$^2$LA still outperforms other methods in terms of overall accuracy, averaging as 0.58%, 0.44% on ImageNet-LT and Places-LT. Specifically, our method provides 0.43%/2.30%/3.50% and 0.58%/0.83%/0.55% improvements in linear probing *w.r.t.* many/medium/few group on ImageNet-LT and Places-LT. The consistent performance overhead indicates the robustness of our method to deal with long-tailed distribution with different characteristics. Moreover, the averaging improvement of standard deviation is 1.50 on ImageNet-LT, indicating the comprehensive merits on the tail classes towards representation balancedness. However, an interesting phenomenon is that the fine-grained performance exhibits a different trend on Places-LT. We observe that the standard deviation of our method does not significantly decrease on Places-LT. A possible explanation is that the head classes suffer from a severer over-expansion in the scene-centric scenario. As can be seen, the performance of head classes is even worse than that of tail classes, which requires more effort and exploration for improvement alongside S$^2$LA.

Table 3: Supervised long-tailed learning by finetuning on CIFAR-100-LT-R100 and ImageNet-LT. We compare the performance of four self-supervised learning methods as the pre-training stage for downstream supervised logit adjustment (Menon et al., 2021) method. Besides, the performance of logit adjustment via learning from scratch is also reported for comparisons.

| Dataset | LA | Logit adjustment pretrained with the following SSL methods | | | | | | | | | | |
|---|---|---|---|---|---|---|---|---|---|---|---|---|
| | | SimCLR | +$S^2$LA | **Improv.** | Focal | +$S^2$LA | **Improv.** | SDCLR | +$S^2$LA | **Improv.** | BCL | +$S^2$LA | **Improv.** |
| CIFAR-100-LT | 46.61 | 49.95 | 50.67 | **+0.72** | 49.87 | 50.98 | **+1.11** | 49.88 | 50.69 | **+0.81** | 50.38 | 50.99 | **+0.61** |
| ImageNet-LT | 48.27 | 51.10 | 51.64 | **+0.54** | 51.25 | 51.52 | **+0.27** | 51.01 | 51.33 | **+0.32** | 51.50 | 51.76 | **+0.26** |

## 5.3 DOWNSTREAM SUPERVISED LONG-TAILED CLASSIFICATION

In supervised long-tailed learning, self-supervised learning has been proved to be beneficial as a pre-training stage to exclude the explicit bias from the class imbalance (Yang & Xu, 2020; Liu et al., 2021; Zhou et al., 2022). In order to validate the representation transferability of $S^2$LA, we conduct self-supervised pre-training as the initialization for downstream supervised classification tasks on CIFAR-100-LT-R100 and ImageNet-LT. The state-of-the-art logit adjustment (Menon et al., 2021) is chosen as the baseline. The combination of $S^2$LA + LA is interpreted as a method where $S^2$LA aims at the re-balanced representation extraction and LA copes with the debiased classifier learning. In Table 3, we can find that the superior performance improvements is achieved by self-supervised pre-training over the direct supervised learning baseline. We can also observe that our method consistently outperforms other methods, averaging as 0.81% and 0.35% on CIFAR-100-LT-R100 and ImageNet-LT. These results demonstrate that $S^2$LA are well designed to facilitate long-tailed representation learning and improve the generalization ability for downstream supervised tasks.

## 5.4 FURTHER ANALYSIS AND ABLATION STUDIES

**Optimal Transport.** We conduct a range of experiments to verify the effectiveness of our design, which includes (I-1) the logit-adjusted loss with soft assignments, (I-2) the logit-adjusted loss with hard assignments, (I-3) post-hoc logit-adjusted loss with soft assignments, (I-4) post-hoc logit-adjusted loss with hard assignments. Besides, we also add two methods: (J) joint optimization on $p$, $q$ and (O) oracle ground-truth guidance. From the results, both the logit-adjusted and the post-hoc logit-adjusted manners (I1-I4) lead to performance degradation compared with the vanilla SimCLR, while the label allocation by optimal transport ($S^2$LA) significantly improves the performance.

**Diverse Pre-defined Skewed Constraints.** To further validate $S^2$LA , we alternate the estimated constraints with the fixed imbalanced prior based on the ground-truth label distribution. We explore the effect of the imbalanced prior by adding a temperature factor $T$, which softens the prior to various scale of skewness. As shown in Figure 3(a), we observe that $S^2$LA outperforms all models pretrained with the imbalanced prior. A possible reason is that our method captures the inherent geometric statistics from

Table 4: Linear probing results *w.r.t* different setups for surrogate labels.

| Acronym | SimCLR | J | $S^2$LA | O |
|---|---|---|---|---|
| Accuracy | 50.72 | 2.91 | 53.96 | 56.47 |
| Acronym | I-1 | I-2 | I-3 | I-4 |
| Accuracy | 48.41 | 49.93 | 50.38 | 50.22 |

the embedding space, which is different from the the label space. This is aligned with the empirical results that the model achieves the worst performance with the vanilla ground-truth prior ($T = 1$), whereas achieving the best performance with the prior closest to $S^2$LA ($T = 0.6$). Moreover, we observe that the uniform prior ($T = 0$) adopted in (Asano et al., 2020; Caron et al., 2020) does not show the expected performance on the long-tailed data.

**Embedding Statistics.** In Figure 3(c), we plot the prediction histogram at two stages in the training process, *i.e.*, warm-up stage and the end of training stage. We observe that the adjusted distribution are more imbalanced than the vanilla distribution. Specifically, it shows more samples are assigned to the instance-rich classes with the shrinking of instance-poor classes. This is aligned with the goal to compress the feature span of head classes and encourage the expansion of tail classes.

**Hyper-parameter $\tau$ for $S^2$LA .** We can adjust the coefficient $\tau$ in Eq. 3 to control the direction and strength of $S^2$LA , as shown in Figure 3(b). For each $\tau$, we perform performance evaluation and skewness estimation, providing the inherent characteristics of our method. From the figure, the correlation between the coefficient $\tau$ and the embedding skewness indicates that we can ad-

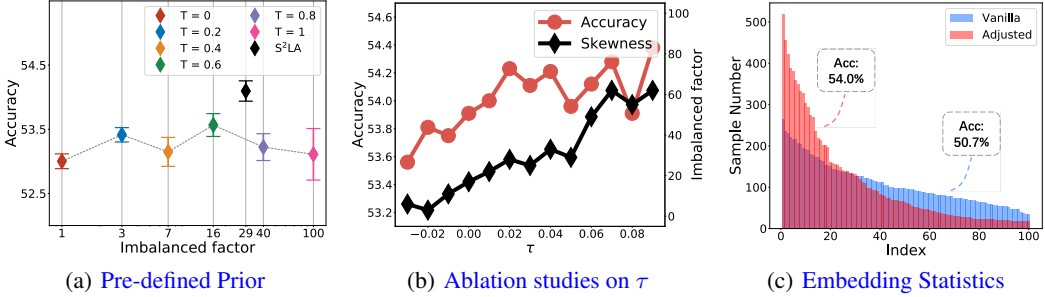

(a) Pre-defined Prior      (b) Ablation studies on $\tau$      (c) Embedding Statistics

Figure 3: (a) Linear probing performance with error bars *w.r.t.* the ground-truth prior served as the categorical constraints. We soften the oracle distribution by $T$ to obtain diverse imbalanced priors and report the corresponding imbalanced factor. (b) Linear probing performance and imbalanced factor of the geometric statistic w.r.t. $\tau$ on CIFAR-100-LT-R100. (c) Histograms of geometric predictions at the end of the warm-up stage (vanilla) or at the end of training (adjusted).

Table 5: Ablation study on the Simplex ETF.

| Method | R100 | R50 | R10 |
|---|---|---|---|
| SimCLR | 50.72 | 52.24 | 55.67 |
| +S$^2$LA | 53.96 | 55.42 | 57.36 |
| +S$^2$LA(RC) | 53.41 | 54.78 | 56.54 |
| +Simplex ETF | 51.10 | 51.99 | 55.56 |

Table 6: The time cost of mini-batch training (seconds) on CIFAR-100-LT, ImageNet-LT and Places-LT.

| | CIFAR-LT | | ImageNet-LT | | Places-LT | |
|---|---|---|---|---|---|---|
| SimCLR(+S$^2$LA) | 0.379 | 0.407 | 0.757 | 0.787 | 0.724 | 0.747 |
| Focal(+S$^2$LA) | 0.423 | 0.467 | 0.943 | 1.008 | 1.001 | 1.050 |
| SDCLR(+S$^2$LA) | 0.374 | 0.397 | 0.752 | 0.771 | 0.759 | 0.776 |
| BCL(+S$^2$LA) | 0.377 | 0.405 | 0.756 | 0.783 | 0.722 | 0.745 |

just the skewness flexibly. Considering $\tau = 0$ as an anchor point, we can achieve the promising performance with a proper strength in the process of increasing skewness. In contrast, the model performance degenerates when we reversely adopt S$^2$LA (decrease skewness). These observations further confirm that our method can realize any desired skewness to deal with different levels of data imbalancedness.

**Ablation Study on Simplex ETF.** To further verify the effectiveness of Simplex ETF, we conduct experiments to investigate our methods with the random linear classifier. In Table 5, we can see that the random classifier as a measure under the framework of S$^2$LA does improve the performance of SimCLR, but fails to outperforms Simplex ETF. Besides, we also conduct experiments with Simplex ETF as the projector. As can be seen, if Simplex ETF alone is used to balance the representation learning, the improvement is minor and sometimes degrades. This is because the direct estimation from the Simplex is noisy during training when the representation is not ideally distributed.

**Computational cost.** In Table 6, we provide the mini-batch training time of different methods on CIFAR-100-LT, ImageNet-LT and Places-LT. In our runs, the proposed S$^2$LA incurs an average 7.8%, 4.2%, 3.4% computational overhead on CIFAR-100-LT, ImageNet-LT and Places-LT, respectively, which is relatively lightweight compared to the computational cost of deep neural networks.

## 6 CONCLUSION

In this paper, we study why the conventional contrastive learning loss underperforms in self-supervised long-tailed learning, motivating our exploration on category-level uniformity instead of sample-level uniformity. Inspired by the power of logit adjustment in supervised long-tailed learning, we correspondingly propose a Self-Superivsed Logit Adjustment algorithm to calibrate the learning in the self-supervised learning paradigm. On the geometric level, S$^2$LA gradually rectify the feature span by an alternation between the logit adjustment and the surrogate label allocation. Moreover, S$^2$LA is orthogonal to existing self-supervised long-tailed methods and can be easily plugged into these methods in an efficient manner. Extensive experiments demonstrate the consistent efficacy of our proposed S$^2$LA. We believe that the geometric perspective has the more potential to understand self-supervised learning methods, especially when coping with the long-tailed scenarios. In the future, we will extend our work with several potential directions such as theoretical definition of the geometric structure over the training embeddings, and investigations of other clustering methods or regularization to edit the geometric statistics.

## 7 ETHICS STATEMENT

This paper does not raise any ethics concerns. This study does not involve any human subjects, practices to data set releases, potentially harmful insights, methodologies and applications, potential conflicts of interest and sponsorship, discrimination/bias/fairness concerns, privacy and security issues, legal compliance, and research integrity issues.

## 8 REPRODUCIBILITY STATEMENT

To ensure the reproducibility of experimental results, we will provide the anonymous repository about our codes in the discussion phase for reviewing purposes. The experimental setups for training and evaluation as well as the hyper-parameters are detailed in Section 5.1 and Appendix D.

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

APPENDIX

## A ALGORITHM

We summarize the complete procedure of our $S^2LA$ method in Algorithm 1.

---

**Algorithm 1** Our proposed $S^2LA$.

---

**Input:** dataset $\mathcal{D}$, number of epochs $E$, number of warm-up epochs $E_w$, geometric classifier $\mathbf{M}$, a self-supervised learning method $\mathcal{A}$, logit adjustment temper $\tau$
**Output:** pretrained model parameter $\theta_E$
**Initialize:** model parameter $\theta_0$

1: Warm up model $\theta$ for $E_w$ epochs according to $\mathcal{A}$.
2: **for** epoch $e = E_w, E_w + 1, \ldots, E$ **do**
3:     Obtain the geometric prediction $\boldsymbol{t} = \arg\max_i h_i^m(\boldsymbol{x})$ for dataset $\mathcal{D}$.
4:     Compute the surrogate class prior $p(\boldsymbol{t})$.
5:     Compute $p\left(\tilde{\boldsymbol{t}}\right)$ using logit adjustment according to Eq. 3.
6:     **for** mini-batch $k = 1, 2, \ldots, B$ **do**
7:         Update the output logits $h^m(\boldsymbol{x})$ for each sample.
8:         Obtain the label $q(\boldsymbol{t}|\boldsymbol{x})$ using the optimal transport solutions according to Eq. 6.
9:         Compute $\mathcal{L}_{\text{SSL}}$ according to $\mathcal{A}$ and the proposed $\mathcal{L}_{S^2LA}$ according to Eq. 4.
10:        Uptate model $\theta$ by minimizing $\mathcal{L}_{\text{SSL}} + \mathcal{L}_{S^2LA}$.
11:     **end for**
12: **end for**

---

## B RELATED WORKS: SUPERVISED LONG-TAILED LEARNING

As the explorations on the classifier learning are orthogonal to the self-supervised learning paradigms, we mainly focus on the representation learning in supervised long-tailed recognition. The pioneering work (Kang et al., 2019) first explored representation and classifier learning with a disentangling mechanisms and showed the merits of instance-balanced sampling strategy on the representation learning stage. Subsequently, (Yang & Xu, 2020) pointed out the negative impact of label information and proposed to improve the representation learning with semi-supervised learning and self-supervised learning. This motivates a stream of research works diving into the representation learning. Supervised contrastive learning (Kang et al., 2020; Cui et al., 2021) is leveraged with rebalanced sampling or prototypical learning design to pursue a more balanced representation space. (Li et al., 2021) explicitly regularizes the class centers to a maximum separation structure with similar drives to the balanced feature space.

## C ON CLASS-BALANCED DATA

According to the proof in (Wang & Isola, 2020), conventional contrastive learning targets to pursue the sample-level uniformity. Given the balanced feature subspace, we can naturally obtain the uniform predictions $p(\boldsymbol{t}) = \frac{1}{K}$ based on Simplex ETF, and the logit adjustment will not modify the distribution skewness as $\tilde{\boldsymbol{t}} = \arg\max p(\boldsymbol{t}|\boldsymbol{x})p(\boldsymbol{t})^\tau = \arg\max p(\boldsymbol{t}|\boldsymbol{x}) = \boldsymbol{t}$. In this case, it is reconcilable between contrastive learning and the second term in Eq.(7), only if optimal transport makes the reallocated labels of samples certain and uniformly distributed regarding the categories. The loss term $-q(\boldsymbol{t}|\boldsymbol{x})\log p(\boldsymbol{t}|\boldsymbol{x})$ will then make negligible effect on the representation learning and the contrastive learning domainates the optimization.

Empirically, we conduct experiments in the balanced setting shown in Table 7. From the results, we can see that $S^2LA$ shows comparable performance with the baseline methods in the balanced setting. This observation is consistent with the aforementioned theoretical analysis.

Table 7: Linear probing results in the balanced setting on CIFAR-100-LT-R100.

| Method | SimCLR | +S$^2$LA | Focal | +S$^2$LA | SDCLR | +S$^2$LA | BCL | +S$^2$LA |
|---|---|---|---|---|---|---|---|---|
| Accuracy | 66.75 | 66.41 | 66.42 | 66.79 | 65.96 | 66.17 | 69.16 | 69.33 |

# D EXPERIMENTAL DETAILS

## D.1 DATASET STATISTICS

We conduct experiments on three benchmark datasets for long-tailed learning, including CIFAR-100-LT (Cao et al., 2019), ImageNet-LT (Liu et al., 2019) and Places-LT (Liu et al., 2019). For small-scale datasets, we adopt the widely-used CIFAR-100-LT with the imbalanced factor of 100, 50 and 10 (Cao et al., 2019).

In Table 8, we summarize the benchmark datasets used in this paper. Long-tailed versions of CIFAR-100 are constructed following the exponential distribution. For large-scale datasets, ImageNet-LT (Liu et al., 2019) has 115.8K images with 1000 categories, ranging from 1,280 to 5 in terms of class cardinality and Places-LT (Liu et al., 2019) contains 62,500 images with 365 categories, with the sample number per category ranging from 4,980 to 5. The large-scale datasets follow Pareto distribution.

Table 8: Statistics of long-tailed datasets. Exp represents exponential distribution.

| Dataset | # Class | Type | Imbalanced Ratio | # Train data | # Test data |
|---|---|---|---|---|---|
| CIFAR-100-LT-R100 | 100 | Exp | 100 | 10847 | 10000 |
| CIFAR-100-LT-R50 | 100 | Exp | 50 | 12608 | 10000 |
| CIFAR-100-LT-R10 | 100 | Exp | 10 | 19573 | 10000 |
| ImageNet-LT | 1000 | Pareto | 256 | 115846 | 50000 |
| Places-LT | 365 | Pareto | 996 | 62500 | 36500 |

## D.2 LINEAR PROBING STATISTICS ON LARGE-SCALE DATASET

The 100-shot evaluation follows the setting in previous works (Jiang et al., 2021; Zhou et al., 2022). As shown in Table 9, full-shot evaluation requires 10x - 30x the amount of data compared with the pre-training dataset, which might not be very practical. In contrast, the scale of 100-shot data is consistent with the pre-training dataset. We also present full-shot evaluation in Section E.5.

Table 9: Statistics of linear probing on large-scale dataset.

| Dataset | # Class | # Training data | # 100-shot data | # full-shot data | # Test data |
|---|---|---|---|---|---|
| ImageNet-LT | 1000 | 115,846 | 100,000 | 1,261,167 | 50,000 |
| Places-LT | 365 | 62,500 | 36,500 | 1,803,460 | 36,500 |

## D.3 IMPLEMENTATION DETAILS

**Toy Experiments.** (Figure 1) We use a 2-Layer ReLU network with 20 hidden units and 2 output units for visualization. The SimCLR algorithm (Chen et al., 2020) is adopted in the warm-up stage with proper Gaussian noise as augmentation. After the warm-up stage, we train S$^2$LA according to Eq. 7. Due to the dimensional constraints, we use the orthogonal classifier [(1,1),(-1,1),(-1,-1),(1,-1)] as the suboptimal structure for maximum separation.

**Linear Probing Evaluation.** We follow (Zhou et al., 2022) to conduct Adam optimizer for 500 epochs based on batch size 128, weight decay factor $5 \times 10^{-6}$ and the learning rate decaying from

$10^{-2}$ to $10^{-6}$. For few-shot evaluation on ImageNet-LT and Places-LT, we use the same sampled 100-shot subsets in (Zhou et al., 2022).

**Computing Infrastructure.** Our codes are built on PyTorch. We trained all the experiments on NVIDIA GeForce RTX 3090.

### D.4 FOCAL LOSS

Focal loss (Lin et al., 2017) is discussed and compared in (Jiang et al., 2021; Zhou et al., 2022) in the context of self-supervised long-tailed learning. Specifically, we use the term inside $\log(\cdot)$ of SimCLR loss as the likelihood to replace the probabilistic term of the supervised Focal loss and obtain the self-supervised Focal loss as:

$$\mathcal{L}_{focal} = -\frac{1}{|\mathcal{B}|}\sum_{b\in\mathcal{B}}\frac{1}{|\mathcal{X}_b|}\sum_{\boldsymbol{x}\in\mathcal{X}_b}(1-\boldsymbol{p})^{\gamma_f}\log(\boldsymbol{p}), \quad \boldsymbol{p} = \frac{\exp\left(f_\theta(\boldsymbol{x})^\top f_\theta(\boldsymbol{x}^+)/\gamma\right)}{\sum_{x^-\in\mathcal{X}_b^-\cup\{\boldsymbol{x}^+\}}\exp\left(f_\theta(\boldsymbol{x})^\top f_\theta(\boldsymbol{x}^-)/\gamma\right)}$$

where $\gamma_f$ is a temperature factor. We defaultly set $\gamma_f$ as 2 in all experiments.

## E MORE COMPREHENSIVE RESULTS

### E.1 ABLATION STUDY ON OPTIMAL TRANSPORT

In the following, we conduct a range of experiments to verify the effectiveness of our design, which includes (I-1) the logit-adjusted loss with soft assignments, (I-2) the logit-adjusted loss with hard assignments, (I-3) Post-hoc logit-adjusted loss with soft assignments. (I-4) Post-hoc logit-adjusted loss with hard assignments. Besides, we also add two methods: joint optimization on $p$, $q$ and oracle ground-truth guidance.

Table 10: Ablation study on optimal transport.

| Method | Acronym | Formulation | Accuracy |
|---|---|---|---|
| SimCLR | - | - | 50.72 |
| +Joint Optimization | - | $\min_{p,q} -\frac{1}{|\mathcal{D}|}\sum_{x\sim\mathcal{D}} q(\boldsymbol{t}|\boldsymbol{x})\log p(\boldsymbol{t}|\boldsymbol{x})$ | 2.91 |
| +Logit-adjusted softmax (soft) | I-1 | $\min_p -\frac{1}{|\mathcal{D}|}\sum_{x\sim\mathcal{D}} p(\boldsymbol{t}|\boldsymbol{x})\log p(\boldsymbol{t}|\boldsymbol{x})/p(\boldsymbol{t})^\tau$ | 48.41 |
| +Logit-adjusted softmax (hard) | I-2 | $\min_p -\frac{1}{|\mathcal{D}|}\sum_{x\sim\mathcal{D}} \boldsymbol{t}\log p(\boldsymbol{t}|\boldsymbol{x})/p(\boldsymbol{t})^\tau$ | 49.93 |
| +Post-hoc (soft) | I-3 | $\min_p -\frac{1}{|\mathcal{D}|}\sum_{x\sim\mathcal{D}} p(\tilde{\boldsymbol{t}}|\boldsymbol{x})\log p(\boldsymbol{t}|\boldsymbol{x})$ | 50.38 |
| +Post-hoc (hard) | I-4 | $\min_p -\frac{1}{|\mathcal{D}|}\sum_{x\sim\mathcal{D}} \tilde{\boldsymbol{t}}\log p(\boldsymbol{t}|\boldsymbol{x})$ | 50.22 |
| +S²LA | - | $\min_{p,q} -\frac{1}{|\mathcal{D}|}\sum_{x\sim\mathcal{D}} q(\boldsymbol{t}|\boldsymbol{x})\log p(\boldsymbol{t}|\boldsymbol{x})$ | 53.96 |
| +Oracle | - | $\min_p -\frac{1}{|\mathcal{D}|}\sum_{(x,y)\sim\mathcal{D}} \boldsymbol{y}\log p(\boldsymbol{t}|\boldsymbol{x})$ | 56.47 |

Table 11: Normalized mutual information (NMI) score between the geometric predictions and the oracle labels.

| Prediction | $\tilde{\boldsymbol{t}}$ with Post-hoc (CE) | $q(\boldsymbol{t}|\boldsymbol{x})$ with S²LA |
|---|---|---|
| NMI score | 0.056 | 0.432 |

As can be seen, both the logit-adjusted losses and the post-hoc logit-adjusted manners (I1-I4) lead to performance degradation compared with the vanilla SimCLR, while the label allocation by optimal transport (S²LA) significantly improves the performance. To explain this, we compute the NMI score between $\tilde{\boldsymbol{t}}$ and the oracle label, and the NMI score between $q(\boldsymbol{t}|\boldsymbol{x})$ and the oracle label, summarized in Table 11. Accorrding to the comparison of their NMIs, we can understand the roughly adjusted $\tilde{\boldsymbol{t}}$ can be noisy and shows the weak correlation with the oracle label, while $q(\boldsymbol{t}|\boldsymbol{x})$ reallocated by optimal transport based on the population statistic of $\tilde{\boldsymbol{t}}$ effectively improves the correlation.

## E.2 ON DIMENSION OF MAXIMUM SEPARATION STRUCTURE

For Simplex ETF, there is a hard dimension constraint in Eq. 2, *i.e.*, $d < K$. However, if this constraint violates, we do not have such a structure in the hyperspherical space. Alternatively, we can conduct the gradient descent to find an approximation of the maximum separation prototypes applied into S$^2$LA. This refers to minimising the following loss function as demonstrated in (Li et al., 2022).

$$\mathcal{L}_{\text{uniform}} = \log \sum_i^K \sum_j^K e^{\tilde{\boldsymbol{m}}_i \cdot \tilde{\boldsymbol{m}}_j / \tau_{\text{u}}}, \quad \text{s.t.} \quad \sum_i^K \tilde{\boldsymbol{m}}_i = 0 \text{ and } \forall i \in K \ \|\tilde{\boldsymbol{m}}_i\| = 1 \tag{8}$$

where the first term penalizes the pairwise similarity of different prototypes (Wang & Isola, 2020) under the constraints of Simplex ETF. Figure 4 shows the performance of S$^2$LA equipped with different dimension of the geometric structure by analytical and approximate ways. Comparing the results of Simplex ETF with those of proxy weights, we can see that the comparable performance is achieved in both geometric structures. This indicates that our method is effective to two forms of the geometric structure, relaxing the hard dimensional constraints in Simplex ETF.

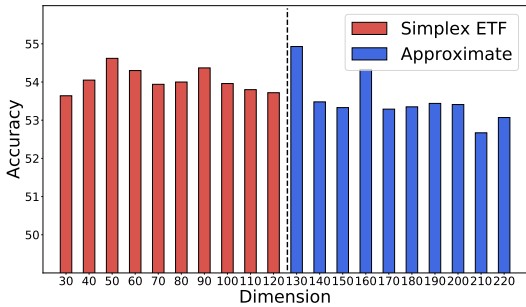

Figure 4: Linear probing performance w.r.t. the dimensionality $d$ of the pre-defined classifier on CIFAR-100-LT-R100. We use simplex ETF structure when $d < K$ or otherwise the approximation learned via Eq. 8.

## E.3 CLUSTERING QUALITY

We track the normalized mutual information (Strehl & Ghosh, 2002) of the geometric predictions with the ground-truth label in the training stage on CIFAR-100-LT. As shown in Figure 5, we observe that S$^2$LA significantly improves the NMI score across different methods, indicating that the distorted embedding space are calibrated to better capture the semantic information. Moreover, we find that the NMI improvements of the existing works are marginal compared with S$^2$LA, which verifies the importance of calibrating the distorted embedding space.

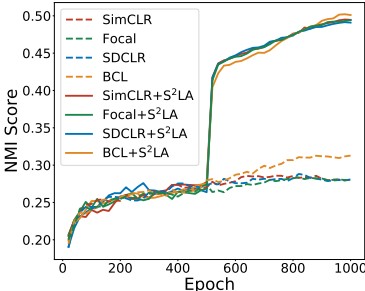

Figure 5: Normalized mutual information (NMI) between the geometric predictions and the ground-truth labels throughout training.

### E.4 COMPARISONS WITH SELA

In this part, we conduct empirical comparisons with SeLa (Asano et al., 2020). In the following, we use the official code of SeLa with the default training settings (marked as SeLa), with our training schedule (marked as SeLa*) and conduct the experiments on CIFAR-100-LT-R100. From the results, we can see that our method can improve the performance of SeLa.

Table 12: Linear probing results of SeLa on CIFAR-100-LT-R100.

| Method | SeLa | SeLa* | SeLa* + $S^2LA$ |
|---|---|---|---|
| Accuracy | 44.45 | 46.47 | 48.10 |

### E.5 FULL-SHOT EVALUATION ON LARGE-SCALE DATASET

Here we provide 100-shot evaluation and full-shot evaluation on ImageNet-LT, as shown in Table 13. We observe that the performance improvements and representation balancedness (Std) are consistent with the 2 evaluations, indicating the rationality of the 100-shot evaluation.

Table 13: Full-shot evaluation and 100-shot evaluation on ImageNet-LT.

| Evaluation | Method | Many | Medium | Few | Std | Avg |
|---|---|---|---|---|---|---|
| 100-shot | SimCLR | 41.69 | 33.96 | 31.82 | 5.19 | 36.65 |
|  | $+S^2LA$ | 41.53 | 36.35 | 35.84 | 3.15 | 38.28 |
| Full-shot | SimCLR | 42.86 | 35.17 | 33.13 | 5.13 | 37.86 |
|  | $+S^2LA$ | 44.11 | 38.59 | 37.87 | 3.41 | 40.62 |

