# OpenReview forum: "Self-Supervised Logit Adjustment"
_ICLR.cc/2023/Conference — Submitted to ICLR 2023_

### Official Review · Reviewer_bvPJ · 2022-10-20

**Confidence:** 4
**Correctness:** 3
**Technical Novelty And Significance:** 4
**Empirical Novelty And Significance:** 2
**Recommendation:** 6

**Clarity, Quality, Novelty And Reproducibility:**

- Clarity: can be improved in the method part.
- Quality: good.
- Originality: good.
- Reproducibility: good, code is promised to be provided.

**Strength And Weaknesses:**

Strength:

- The paper presents a strong motivation, which is to achieve class-level feature uniformity instead of instance-level feature uniformity. The overall method is closely built around this motivation. The toy example in Figure 1. is a good illustration of the paper's idea. The idea is new to the field.

- The empirical results are sufficiently strong. Improvements are observed on multiple datasets (CIFAR-LT, ImageNet-LT, and Places-LT) on top of several SSL methods.


Weakness:

- I find the method part convoluted:

  - I have difficulty understanding the importance of Simplex ETF. To my understanding, Simplex ETF is a fixed linear classifier that outperforms the standard learnable linear classifiers in end-to-end __supervised__ imbalanced learning. However, why is it necessary in the unsupervised setting? Should other alternatives to "measure the geometric characteristics", e.g., a fixed random linear classifier, be discussed and compared?

  - The motivation for logit adjustment is not very clear. Why is logit adjustment required on top of Simplex ETF? "All vectors in a Simplex ETF have the same pair-wise angle", is Simplex EFT alone sufficient to ensure the class-level uniformity already?

  -  Why is Eqn. (4) the objective? Is it to find some surrogate labels $q(t|x)$ to compute the cross-entropy loss with $p(t|x)$, so that the marginal $p(t)$ can be adjusted to $p(\tilde{t})$? This part may need more clarity.

  - The notions are not clear sometimes. Should the expression $p(\tilde{t}|x) \propto p(t|x)p(t)^{1+\tau}$ be $p(\tilde{t}|x) \propto p(\tilde{t}|x)p(\tilde{t})^{1+\tau}$? $q$ is also not defined in Eqn. (4) until later.

  - __Overall__, I believe that the mixing of statistical interpretations (logit adjustment, Bayes) and geometric interpretations (Simplex ETF, geometric span) is what makes the method part hard to understand. Currently, I don't see how the two interpretations can be linked, i.e., the relation between "skewed geometric distribution" and the "skewed statistical distribution". If the authors are just trying to "shrink the feature span of head classes for the category-level uniformity", I would suggest the authors stick to the geometric interpretations and view "logit adjustment" as a margin loss.

- I may have missed the part about how focal loss is used for self-supervised learning. It would be better if more related works besides Lin et al. 2017  can be provided.






**Summary Of The Paper:**

This paper proposed to solve the self-supervised imbalanced learning by enforcing category-level uniformity instead of sample-level uniformity.

**Summary Of The Review:**

The idea of the paper is new and comes with a clear motivation, and the experiment results seem sufficiently strong. However, I find the method part of the paper very difficult to understand which heavily affects my assessment of its correctness. I believe that the method part can be clearer with more polishing efforts, but based on the current version, I would recommend a weak rejection.

---

> ### Author Response · Authors · 2022-11-16
> **Response to Reviewer bvPJ[2/2]**
>
> > **Q3**:  Explanations of Eq. (4).
>
> **A3**: Thanks for the questions. As the reviewer's understanding, the process of optimizing $q(\boldsymbol{t}|\boldsymbol{x})$ is to reallocate the labels with the adjusted statistics $p(\tilde{\boldsymbol{t}})$. The generation quality is guaranteed with the optimal transport on the population level. After inference by optimal transport, the generated $q(\boldsymbol{t}|\boldsymbol{x})$ acts as the supervision  to guide the model outputs $p(\boldsymbol{t}|\boldsymbol{x})$ with cross-entropy loss. The marginals of geometric predictions are then adjusted from $p(\boldsymbol{t})$ to $p(\tilde{\boldsymbol{t}})$ in this stage.
>
>
> > **Q4**: Explanations of some notations.
>
> **A4**: We follow the reviewer's suggestion to refine the description, and please refer to the revision. Regarding the reviewer's question, we first has $p(\boldsymbol{t}|\boldsymbol{x}) \propto p(\boldsymbol{x}|\boldsymbol{t})p(\boldsymbol{t})$ based on Bayes Theorem, and then along with Eq. (3), we can derive $p\left(\tilde{\boldsymbol{t}}|\boldsymbol{x}\right) \propto p(\boldsymbol{x}|\boldsymbol{t})p(\boldsymbol{t})^{1+\tau}$ with the adjusted term $p(\boldsymbol{t})^{\tau}$.
>
> > **Q5**: Understandings of the proposed methods.
>
> **A5**: Thanks for the reviewer's suggestion. The statistical interpretations and the geometric interpretations are inherently integrated as we manipulate skewness on the predictions of Simplex ETF. On the geometric level, we leverage Simplex ETF predictions $p(\boldsymbol{t}|\boldsymbol{x})$ to determine the typical location of each sample on the hypersphere. On the statistical level, we resort to logit adjustment for approaching the desired marginals $p(\tilde{\boldsymbol{t}})$ with the goal to achieve category-level uniformity. The proposed method motivates a population-level label allocation on top of the geometric classifier with calibrated marginal constraints, counteracting the negative affect of long-tailed data distribution on the representation learning. We will refine the expression to explain the most things from the geometric interpretations to make it more clear and avoid the confusion.
>
> > **Q6**: Related works about Focal loss.
>
> **A6**: Thank you for the suggestion. We have added the related works and more explanations about Focal loss (See Appendix D.4) in the revision.

---

> ### Author Response · Authors · 2022-11-16
> **Response to Reviewer bvPJ[1/2]**
>
> Thank you for your time devoted to reviewing this paper and your constructive suggestions. Here are our detailed replies to your questions.
>
> > **Q1**:  Effectiveness of Simplex ETF classifier.
>
> **A1**: Our motivation is to make self-supervised learning robust to the class-imbalance data, which requires the pursuit in the embedding space intrinsically switching from the sample-level uniformity to the category-level uniformity. The Simplex ETF is a tool to measure the gap between the category-level uniformity and the sample-level uniformity, which is then transformed as the supervision feedback to the training. We have comprehensively modified the submission to make them more clear.
>
> However, we do appreciate the reviewer's advice and conduct experiments to investigate our methods with the random linear classifier in the following table. According to the results, we can see that the random classifier as a measure under the framework of $S^2LA$ does improve the performance of SimCLR, but fails to outperforms Simplex ETF. We have added these results into the draft to clarify the advantages on the choice of Simplex ETF.
>
> [**Table 1.** $S^2LA$ with random classifier on CIFAR-100-LT.]
> | Method             | R100  | R50   | R10   |
> |-----------------------------|:------------:|:-----:|:-----:|
> | SimCLR                      | 50.72 | 52.24 | 55.67 |
> | +$S^2LA$ (Simplex ETF)       | 53.96 | 55.42 | 57.36 |
> | +$S^2LA$ (Random Classifier) | 53.41 | 54.78 | 56.54 |
>
> > **Q2**:  Motivation of logit adjustment.
>
> **A2**: We use an experiment to clarify this point. As shown in the below table, if Simplex ETF alone is used to balance the representation learning, the improvement is minor and sometimes degrades. This is because the direct estimation from the Simplex is noisy during training when the representation is not ideally distributed, while logit adjustment here is to strengthen the estimation of samples from head classes so that they can be discovered as many as possible. Then, the feature regime of head samples can be sufficiently suppressed via Eq.(7) to pursue the category-level uniformity.
>
> We very appreciate the constructive advice of the reviewer and have added ablation experiments on the measure of the geometric statistics to make it more clear.
>
> [**Table 2.** Ablation study on the Simplex ETF Classifier.]
> | Dataset |     R100     |   R100   |  R50  |  R50    |  R10  |   R10    |
> |----------------|:------------:|------|:-----:|------|:-----:|------|
> |   |      Acc     |  Std |  Acc  |  Std |  Acc  |  Std |
> | SimCLR         |     50.72    | 3.82 | 52.24 | 3.30 | 55.67 | 1.98 |
> | +$S^2LA$        |     53.96    | 2.99 | 55.42 | 3.09 | 57.36 | 1.75 |
> | +Simplex ETF   | 51.10        | 3.67 | 51.99 | 3.23 | 55.56 | 1.93 |

---

> ### Author Response · Authors · 2022-11-17
> **Looking forward to your response**
>
> Dear Reviewer bvPJ,
>
> We sincerely thank you for your great efforts in the review of our manuscript. As the discussion period is approaching its end, we would be grateful if you could confirm whether our responses and the additions we have made to the draft addressed your concerns, and let us know if any issues remain.
>
> Best,
>
> Authors of Paper 3711

---

> ### Author Response · Authors · 2022-11-18
> **Would you mind checking our response? welcome for more discussions.**
>
> Dear Reviewer bvPJ,
>
> We sincerely thank you for your great efforts in the review of our manuscript! As the Stage 1 Discussion period is approaching its end, here is a summary of our previous response and update:
>
>
> - Revised the motivation of Simplex ETF (see Section 3.3, Section 4) with more empirical evidence (see Section 5.4).
> - Enriched the discussion about logit adjustment (see Section 4) and added supportive results (see Appendix 5.4).
> - Expanded the discussion on optimal transport (see Section 4) with more empirical evidence (see Section 5.4).
> - Clarified the understandings of the proposed method.
> - Added explanations about some notations and more details about Focal Loss (see Appendix D.4).
>
> We would be grateful  if you could check our responses with the updated manuscript, and confirm whether our responses have addressed your concerns. Please let us know if there are any further questions or suggestions that we could clarify or improve. More discussions are always welcome!
>
> Best,
>
> Authors of Paper 3711

---

> ### Author Response · Authors · 2022-11-23
> **Looking forward to your response or further suggestions**
>
> Dear Reviewer bvPJ,
>
> We sincerely thank you for your great efforts in the review of our manuscript. We have carefully considered your initial concerns and revised our draft. We are happy to discuss with you if there are remain concerns. New suggestions/comments are also welcome!
>
> Best,
>
> Authors of Paper 3711

---

> > ### Comment · Reviewer_bvPJ · 2022-11-24
> > **Response to Rebuttal**
> >
> > The authors have done an excellent job of showing the effectiveness of SimplexETF and OT-based LA with new experiments in Tables 4&5. From the empirical side, all techniques proposed in the paper can be well supported and finally contribute to a good overall result with acceptable additional training time. After revision, I believe the paper has enough empirical value to benefit future research on self-supervised imbalanced learning, particularly with the code provided. Thus, I raised the score.
> >
> > However, I agree with Reviewer 1EQS that "communication of ideas needs to be improved". I find that the new explanations added in the rebuttal and the revised paper do not fully address my concerns. They are still a bit difficult to understand, e.g., "strengthen the estimation of samples from head classes " as mentioned by Reviewer 1EQS, and "which is then transformed as the supervision feedback to the training.". The explanations do not sound very intuitional or rigorous to me. I appreciate the empirical value of the paper, but I believe that the writing should be polished.

---

> > > ### Author Response · Authors · 2022-11-24
> > > **Thanks for the response**
> > >
> > > Dear Reviewer bvPJ,
> > >
> > > Thank you very much for your positive support and further comments. We will follow the advice to comprehensively refine the descriptions and explanations with more sufficient support in the final version.
> > >
> > > Best,
> > >
> > > Authors of Paper 3711

---

### Official Review · Reviewer_jKEp · 2022-10-23

**Confidence:** 4
**Correctness:** 3
**Technical Novelty And Significance:** 2
**Empirical Novelty And Significance:** 2
**Recommendation:** 5

**Clarity, Quality, Novelty And Reproducibility:**

Clarity: The value and importance of the work seems clear in the current form. The presentation of the proposed method seems understandable. The explanations to the experimental results are sensible. However, the derivation of the objective function in the proposed algorithm needs further interrogation so that the reader understands the technical details.

Quality: The idea and concept of the proposed algorithm look quite plain. It seems that the proposed approach is a straightforward extension of similar approaches. The proposed research seems acceptable in terms of the motivation but lacks significance in theory and experiments.

Originality: Logit adjustment is a reasonable choice and the current paper uses a number of mathematical means to deal with the objective functions. The proposed framework, based on several works with similar drives, has made incremental contribution on solving the SSL problems with the SK algorithm. The originality of the proposed method is very minor.

Reproducibility: There is no indication that the source code and other corresponding materials will be published.

**Details Of Ethics Concerns:**

This proposed research uses publicly accessible datasets so there is no concern on ethics.

**Strength And Weaknesses:**

Strengths -

1. The presentation of the proposed method is generally understandable. The discussion on the theoretical analysis and the experimental is clear.
2. The balance between the proposed approach and the evaluation section look reasonable. Many results have been provided to support the arguments in the theoretical analysis.
3. The mathematical principles and basics have been clearly defined and explained.

Weaknesses -

1. The major concern is that the proposed approach seems incremental, compared to the established works and similar technologies.
2. The evaluations seem to support the arguments that the proposed technology outperforms the other state of the art methods, however, the justification is not comprehensive and requires much additional evidence to support the overall statements.
3. The experimental results show that the proposed algorithm can only achieve slightly better performance (often < 1%) than the other state of the art technologies.

Actions to be taken:

1. Since there is no ground-truth annotation of each sample, and using geometric labels tends to generate trivial unconstrained probabilities, a surrogate label allocation with logit adjustment is used in this paper. Eq. (4) is proposed and then needs to be solved using Eqs. (5)-(6). Finally, the overall objective function is expressed as Eq. (7). However, Eq. (7) has a from similar to that of Zhou et al. (2022). The only difference between them is that the current paper introduces Eq. (4) to its final objective function, which plays as a constraint term. It is not clear if or not this objective function is correctly derived in this paper. If it is, how much difference between the two papers indeed?
2. In the ablation experiments, Fig. 4(c) is not of much value. This is because presenting different distributions without showing the classification results is not convincing.
3. In Fig. 5, increasing skewness also leads to higher accuracy. However, the improvement of accuracy is fairly mild, normally less than 1%. If this is a successful mechanism, the increment should be significant, is not it?

**Summary Of The Paper:**

Given the research progress on logit adjustment for supervised long-tailed learning, this paper presents a new method S^2LA to conduct self-supervised long-tailed learning  from the geometric perspective. In fact, without knowing the class distribution, S^2LA uses a constant simplex ETF to measure the geometric characteristics of the embedding space. Using a surrogate label allocation to represent the target distribution, this paper intends to compress the space of the head classes whilst avoiding the collapse of the tail classes. Treating the above transformation as a balancing and optimal transport problem, the proposed solution is intended to effectively handle the category-level uniformity problem.

**Summary Of The Review:**

In summary, this paper presents an interesting topic, worth of being investigated. The theoretical analysis sounds reasonable but lacks certain explanations to the proposed objective function and its derivation. The experimental results are quite considerate but some of the systematic parameters have not been properly justified in the examination.

---

> ### Author Response · Authors · 2022-11-16
> **Response to Reviewer jKEp**
>
> Thank you for your time devoted to reviewing this paper and your constructive suggestions. Here are our detailed replies to your questions.
>
> > **Q1**:  Novelty of the proposed method.
>
> **A1**: We would like to kindly clarify their differences as follows.
>
> (1) BCL [1] rebalances self-supervised learning from the data perspective, where it captures the information of the historical loss statistic to drive the augmentation for different (head or tail) samples. BCL does not alter the loss forms but controls the augmentation strength. Our method dives into the characteristic of the loss and directly alters the loss to implement the representation balancing. As claimed in the 2nd paragraph of Introduction, they are from two different perspectives.
>
> (2) Two methods are complementary. As demonstrated in Table 1, Table 2 and Table 3, applying our method into BCL [1] consistently improve the performance of self-supervised long-tailed learning and achieves the new state-of-the-art results on these datasets.
>
> > **Q2**:  Additional evidence for Fig. 4\(c\).
>
> **A2**: We appreciate the reviewer's suggestion and have added the classification results to the Fig. 4(c) (3c in the revision).
>
>
> > **Q3**:  Performance improvements.
>
> **A3**: We are sorry to induce the misunderstanding of the reviewer. We would like to kindly clarify that the reported experiments in Figure 5 (3b in the revision) are not the baseline methods, while the baseline SimCLR can only achieve 50.72\% accuracy. The main idea of the ablation study on $\tau$ in Eq. (3) is to demonstrate the consistency between large skewness and better performance in the chosen settings. We will refine the paragraphs in the revision to ease the understanding.
>
> [1] Zhou Z, Yao J, Wang Y F, et al. Contrastive Learning with Boosted Memorization. International Conference on Machine Learning. 2022.

---

> ### Author Response · Authors · 2022-11-17
> **Looking forward to your response**
>
> Dear Reviewer jKEp,
>
> We sincerely thank you for your great efforts in the review of our manuscript. As the discussion period is approaching its end, we would be grateful if you could confirm whether our responses and the additions we have made to the draft addressed your concerns, and let us know if any issues remain.
>
> Best,
>
> Authors of Paper 3711

---

> ### Author Response · Authors · 2022-11-18
> **Would you mind checking our response? welcome for more discussions.**
>
> Dear Reviewer jKEp,
>
> We sincerely thank you for your great efforts in the review of our manuscript! As the Stage 1 Discussion period is approaching its end, here is a summary of our previous response and update:
>
>
> - Clarified the novelty of the proposed method.
> - Enriched the evidence for Figure. 3(c) (see Section 5.4).
> - Clarified the misunderstanding for Figure 3(b) (see Section 5.4).
>
> We would be grateful  if you could check our responses with the updated manuscript, and confirm whether our responses have addressed your concerns. Please let us know if there are any further questions or suggestions that we could clarify or improve. More discussions are always welcome!
>
> Best,
>
> Authors of Paper 3711

---

> ### Author Response · Authors · 2022-11-23
> **Looking forward to your response or further suggestions**
>
> Dear Reviewer jKEp,
>
> We sincerely thank you for your great efforts in the review of our manuscript. We have carefully considered your initial concerns and revised our draft. We are happy to discuss with you if there are remain concerns. New suggestions/comments are also welcome!
>
> Best,
>
> Authors of Paper 3711

---

> ### Author Response · Authors · 2022-11-26
> **We anticipate your feedback!**
>
> Dear Reviewer jKEp,
>
> As the discussion period is approaching its end, we sincerely look forward to your feedback. The authors deeply appreciate your valuable time and efforts spent reviewing this paper and helping us improve it.
>
> It would be very much appreciated if you could once again help review our responses and let us know if these address or partially address your concerns and if our explanations are heading in the right direction.
>
> Please also let us know if there are further questions or comments about this paper. We strive to improve the paper consistently, and it is our pleasure to have your feedback!
>
> Best,
>
> Authors of Paper3711

---

> ### Comment · Reviewer_jKEp · 2022-11-28
> **further comments after rebuttal**
>
> Many thanks the authors for updating their paper with additional work. However, the review comments of the current reviewer are only partially addressed (in particular the methodology), and therefore the review score will remain without change.

---

> > ### Author Response · Authors · 2022-11-28
> > **Thanks for the response**
> >
> > Dear Reviewer jKEp,
> >
> > Thanks for the response. Although the score remains, we appreciate the reviewer's suggestion, and we would be very grateful if you could clarify the unaddressed concerns in the methodology so that we can improve in the future version.
> >
> > Best,
> >
> > Authors of Paper3711

---

> > > ### Author Response · Authors · 2022-12-05
> > > **Would you mind clarifying your concerns? welcome for more discussions.**
> > >
> > > Dear Reviewer jKEp,
> > >
> > > As the discussion period is approaching its end, we would be grateful if you could clarify the unaddressed concerns in the methodology so that we can improve in the future version. New suggestions or comments are also welcome!
> > >
> > > Best,
> > >
> > > Authors of Paper 3711

---

> > ### Author Response · Authors · 2022-12-07
> > **Would you mind clarifying your concerns? welcome for more discussions.**
> >
> > Dear Reviewer jKEp,
> >
> > Thanks again for the response. We have carefully considered your comments to improve our submission. In summary, we have (1) clarified the novelty of the proposed method, (2) enriched the evidence for Figure 3c and (3) clarified the misunderstanding for Figure 3b. Would you mind clarifying the unaddressed concerns for further discussions and improvements of the manuscript. Thank you very much!
> >
> > Best,
> >
> > Authors of Submission 3711

---

> > ### Author Response · Authors · 2022-12-08
> > **Thanks for the response. Looking forward to your detailed feedback or further suggestions.**
> >
> > Dear Reviewer jKEp,
> >
> > We sincerely thank the reviewer for the devoting time and efforts in reviewing the submission. In the following, we summarize the reviewer's comments in the early interaction: (1) novelty of the proposed method, (2) additional evidence for Fig. 3c and (3) performance improvements in Figure 3b.
> >
> > We appreciate the the reviewer's concern and have carefully considered your comments to improve our submission. It would be very much appreciated if you could detail the unaddressed concerns so that we can further clarify and improve our submission in the final revision. We strive to improve the manuscript consistently, and it is our pleasure to have your feedback. Thank you very much!
> >
> > Best,
> >
> > Authors of Submission 3711

---

> > ### Author Response · Authors · 2022-12-09
> > **Thanks for the response. We anticipate your detailed feedback as the deadline is approaching!**
> >
> > Dear Reviewer jKEp,
> >
> > As the discussion period is approaching its end, we sincerely look forward to your detailed feedback. The authors deeply appreciate your valuable time and efforts spent reviewing this paper and helping us improve it.
> >
> > In the following, we summarize the reviewer's comments in the early interaction: (1) novelty of the proposed method, (2) additional evidence for Fig. 3c and (3) performance improvements in Figure 3b.
> >
> > We have carefully considered your comments to improve our submission. It would be very much appreciated if you could detail the unaddressed concerns so that we can further clarify and improve our submission in the final revision.  Thank you very much!
> >
> > Best,
> >
> > Authors of Paper 3711

---

> > ### Author Response · Authors · 2022-12-10
> > **Thanks for the response. Would you mind clarifying your concerns? We anticipate your detailed feedback as the deadline is approaching!**
> >
> > Dear Reviewer jKEp,
> >
> > As the discussion period is approaching its end, we sincerely look forward to your detailed feedback. The authors deeply appreciate your valuable time and efforts spent reviewing this paper and helping us improve it.
> >
> > In the following, we summarize the reviewer's comments in the early interaction: (1) novelty of the proposed method, (2) additional evidence for Figure 3c and (3) performance improvements in Figure 3b.
> >
> > We have carefully considered your comments to improve our submission. In summary, we have (1) clarified the novelty of the proposed method, (2) enriched the evidence for Figure 3c and (3) clarified the misunderstanding for Figure 3b.
> >
> > It would be very much appreciated if you could detail the unaddressed concerns so that we can further clarify and improve our submission in the final revision.  We strive to improve the manuscript consistently, and it is our pleasure to have your feedback. Thank you very much!
> >
> > Best,
> >
> > Authors of Paper 3711

---

### Official Review · Reviewer_EKLC · 2022-10-25

**Confidence:** 4
**Correctness:** 3
**Technical Novelty And Significance:** 3
**Empirical Novelty And Significance:** 3
**Recommendation:** 6

**Clarity, Quality, Novelty And Reproducibility:**

Communication of the ideas can be improved. I don't think I can reproduce results from the paper itself, however authors release the code which somewhat mitigates the issue.

**Strength And Weaknesses:**

Strength:
1. An important problem of SSL on imbalanced classes.
2. An original idea of using surrogate classes to prevent bias due "head" classes.
3. Substantial empirical results.
4. Authors share code for reproducibility.

Weaknesses:
1. Mathematical exposition of the material is poor.
2. Ideas are not communicated clearly.
3. There is no discussion on the technique's limitations.

**Summary Of The Paper:**

The paper addresses an important problem of self-supervised learning on data with imbalanced classes. SSL trained on sample-level uniformity will learn representation for common classes. However, less represented classes will suffer due to bias for "head" classes. The paper introduces a method to achieve category-level uniformity by using surrogate classes. A distinct property of the method is that it can be applied to any SSL method by modifying the loss function along with the training procedure. Authors demonstrate strong empirical results of the method.

**Summary Of The Review:**

After addressing the feedback, communication of ideas somewhat improved. I am giving a conditional acceptance, if authors can add a discussion on the limitations of the approach. Such a discussion will bring forth implicit assumption where the method expected to work.

---

> ### Author Response · Authors · 2022-11-16
> **Response to Reviewer EKLC[2/2]**
>
> > **Q2**:  Assumptions of the proposed method.
>
> **A2**: According to the proof in [1], conventional contrastive learning targets to pursue the sample-level uniformity. Given the balanced feature subspace, we can naturally obtain the uniform predictions $p(\boldsymbol{t}) = \frac{1}{K}$ based on Simplex ETF, and the logit adjustment will not modify the distribution skewness as $\tilde{\boldsymbol{t}} = \arg\max p(\boldsymbol{t}\vert\boldsymbol{x})p(\boldsymbol{t})^{\tau} = \arg\max p(\boldsymbol{t}\vert\boldsymbol{x}) = \boldsymbol{t}$. In this case, it is reconcilable between contrastive learning and the second term in Eq.(7), only if optimal transport makes the reallocated labels of samples certain and uniformly distributed regarding the categories. The loss term $- q(\boldsymbol{t}\vert\boldsymbol{x})\log p(\boldsymbol{t}\vert\boldsymbol{x})$  will than make negliable effect on the representation learning and the contrastive learning domainates the optimization.
>
> To further validate this conjecture and address the reviewer's concern, we conduct experiments in the balanced setting shown in the following table. From the results, we can see that $S^2LA$ shows comparable performance with the baseline methods in the balanced setting. This observation is consistent with the aforementioned theoretical analysis.
>
> [**Table 3.** Ablation study in the balanced setting on CIFAR-100-LT-R100.]
> | Method   | SimCLR | +$S^2LA$  | Focal | +$S^2LA$  | SDCLR | +$S^2LA$  |  BCL  | +$S^2LA$  |
> |----------|:------:|:--------:|:-----:|:--------:|:-----:|:--------:|:-----:|:--------:|
> | Accuracy |  66.75 |   66.41  | 66.42 |   66.79  | 65.96 |   66.17  | 69.16 |   69.33  |
>
> However, we do appreciate the reviewer's advice and enrich the discussion and assumption to clarify where it works or not in the revised submission.
>
>
> > **Q3**:  Support for key intuition.
>
> **A3**: It is hard to visualize the representation space via tSNE in the benchmark datasets like CIFAR100-LT due to too many class numbers. Therefore, we present a Figure 1 on a simulated imbalance dataset to better characterize the dynamic of contrastive learning in terms of the uniformity. We have strengthen the connection between the paragraphs mentioned by the reviewer and Figure 1 to ease the understanding about our intuition.
>
> > **Q4**:  Mathematical definition.
>
> **A4**: We have followed the reviewer's advice to simplify the notation, add more description and refine more paragraphs in the revision to ease the understanding.
>
>
> > **Q5**:  Minor issue.
>
> **A5**: We have removed "anti" to avoid the misunderstanding in the revision.
>
> [1] Wang T, Isola P. Understanding contrastive representation learning through alignment and uniformity on the hypersphere. International Conference on Machine Learning. 2020.

---

> ### Author Response · Authors · 2022-11-16
> **Response to Reviewer EKLC[1/2]**
>
> Thank you for your time devoted to reviewing this paper and your constructive suggestions. Here are our detailed replies to your questions.
>
> > **Q1**:  Explanations of Eq. (4).
>
> **A1**: We very appreciate the advice of the reviewer and have refined the descriptions about Eq.(4) and the notations. In the following, we conduct a range of experiments to verify the effectiveness of our design, which includes (I-1) the logit-adjusted loss with soft assignments, (I-2) the logit-adjusted loss with hard assignments, (I-3) post-hoc logit-adjusted loss with soft assignments, (I-4) post-hoc logit-adjusted loss with hard assignments. Besides, we also add two methods: joint optimization on $p$, $q$ and oracle ground-truth guidance.
>
> [**Table 1.** Ablation study on optimal transport]
> | Method                         | Acronym |                                                                              Formulation                                                                             | Accuracy |
> |--------------------------------|:-------:|:--------------------------------------------------------------------------------------------------------------------------------------------------------------------:|:--------:|
> | SimCLR                         |    -    |                                                                                   -                                                                                  |   50.72  |
> | +Joint Optimization            |    -    |              $\min_{p,q} -\frac{1}{\vert\mathcal{D}\vert} \sum_{x \sim \mathcal{D}}  q(\boldsymbol{t}\vert\boldsymbol{x}) \log p(\boldsymbol{t}\vert\boldsymbol{x})$             |   2.91   |
> | +Logit-adjusted softmax (soft) |   I-1   | $\min_{p} -\frac{1}{\vert\mathcal{D}\vert} \sum_{x \sim \mathcal{D}}  p(\boldsymbol{t}\vert\boldsymbol{x}) \log p(\boldsymbol{t}\vert\boldsymbol{x}) / p(\boldsymbol{t})^{\tau}$ |   48.41  |
> | +Logit-adjusted softmax (hard) |   I-2   |           $\min_{p} -\frac{1}{\vert\mathcal{D}\vert} \sum_{x \sim \mathcal{D}}  \boldsymbol{t} \log p(\boldsymbol{t}\vert\boldsymbol{x})/ p(\boldsymbol{t})^{\tau}$           |   49.93  |
> | +Post-hoc (soft)               |   I-3   |           $\min_{p} -\frac{1}{\vert\mathcal{D}\vert} \sum_{x \sim \mathcal{D}}  p(\tilde{\boldsymbol{t}}\vert\boldsymbol{x}) \log p(\boldsymbol{t}\vert\boldsymbol{x})$          |   50.38  |
> | +Post-hoc (hard)               |   I-4   |                    $\min_{p} -\frac{1}{\vert\mathcal{D}\vert} \sum_{x \sim \mathcal{D}}  \tilde{\boldsymbol{t}} \log p(\boldsymbol{t}\vert\boldsymbol{x})$                    |   50.22  |
> | +$S^2LA$                       |    -    |              $\min_{p,q} -\frac{1}{\vert\mathcal{D}\vert} \sum_{x \sim \mathcal{D}}  q(\boldsymbol{t}\vert\boldsymbol{x}) \log p(\boldsymbol{t}\vert\boldsymbol{x})$             |   53.96  |
> | +Oracle                        |    -    |                      $\min_{p} -\frac{1}{\vert\mathcal{D}\vert} \sum_{(x,y) \sim \mathcal{D}}  \boldsymbol{y} \log p(\boldsymbol{t}\vert\boldsymbol{x})$                      |   56.47  |
>
> [**Table 2.** Normalized mutual information (NMI) between the geometric predictions and the oracle labels]
> | Prediction | $\tilde{\boldsymbol{t}}$ with Post-hoc (CE) | $q(\boldsymbol{t} \vert \boldsymbol{x})$ with $S^2LA$  |
> |------------|:--------------------------------:|:--------------------------:|
> | NMI score  |               0.056              |            0.432           |
>
> As can be seen, both the logit-adjusted losses and the post-hoc logit-adjusted manners (I1-I4) lead to performance degradation compared with the vanilla SimCLR. Specially, if we use $p(\tilde{\boldsymbol{t}}|\boldsymbol{x})$ ($p(\tilde{\boldsymbol{t}})$ is class-level and thus not appliable), namely I-3, the performance has a significant gap with that we use the label allocation by optimal transport ($S^2LA$ ). To explain this, we compute the NMI score between $\tilde{\boldsymbol{t}}$ and the oracle label, and the NMI score between $q(\boldsymbol{t}|\boldsymbol{x})$ and the oracle label, summarized in the below table. Accorrding to the comparison of their NMIs, we can understand the roughly adjusted $\tilde{\boldsymbol{t}}$ can be noisy and shows the weak correlation with the oracle label, while $q(\boldsymbol{t}|\boldsymbol{x})$ reallocated by optimal transport based on the population statistic of $\tilde{\boldsymbol{t}}$ effectively improves the correlation.
>
> We have added these results and discussions into the draft to support the intution behind Eq. (4).

---

> ### Author Response · Authors · 2022-11-17
> **Looking forward to your response**
>
> Dear Reviewer EKLC,
>
> We sincerely thank you for your great efforts in the review of our manuscript. As the discussion period is approaching its end, we would be grateful if you could confirm whether our responses and the additions we have made to the draft addressed your concerns, and let us know if any issues remain.
>
> Best,
>
> Authors of Paper 3711

---

> ### Author Response · Authors · 2022-11-18
> **Would you mind checking our response? welcome for more discussions.**
>
> Dear Reviewer EKLC,
>
> We sincerely thank you for your great efforts in the review of our manuscript! As the Stage 1 Discussion period is approaching its end, here is a summary of our previous response and update:
>
>
> - Expanded the discussion on optimal transport (see Section 4) with more empirical evidence (see Section 5.4).
> - Enriched the discussion and assumption to clarify where our method works and added supportive results (see Appendix C).
> - Clarified the intuition for logit adjustment (see Section 4).
> - Carefully refined the mathematical definition (see Section 4).
>
> We would be grateful  if you could check our responses with the updated manuscript, and confirm whether our responses have addressed your concerns. Please let us know if there are any further questions or suggestions that we could clarify or improve. More discussions are always welcome!
>
> Best,
>
> Authors of Paper 3711

---

> ### Author Response · Authors · 2022-11-23
> **Looking forward to your response or further suggestions**
>
> Dear Reviewer EKLC,
>
> We sincerely thank you for your great efforts in the review of our manuscript. We have carefully considered your initial concerns and revised our draft. We are happy to discuss with you if there are remain concerns. New suggestions/comments are also welcome!
>
> Best,
>
> Authors of Paper 3711

---

> ### Author Response · Authors · 2022-11-26
> **We anticipate your feedback!**
>
> Dear Reviewer EKLC,
>
> As the discussion period is approaching its end, we sincerely look forward to your feedback. The authors deeply appreciate your valuable time and efforts spent reviewing this paper and helping us improve it.
>
> It would be very much appreciated if you could once again help review our responses and let us know if these address or partially address your concerns and if our explanations are heading in the right direction.
>
> Please also let us know if there are further questions or comments about this paper. We strive to improve the paper consistently, and it is our pleasure to have your feedback!
>
> Best,
>
> Authors of Paper3711

---

> ### Author Response · Authors · 2022-12-05
> **Would you mind checking our response? welcome for more discussions.**
>
> Dear Reviewer EKLC,
>
> As the discussion period is approaching its end, we would be grateful if you could check our responses with the updated manuscript, and confirm whether our responses have addressed your concerns. Please let us know if there are any further questions or suggestions that we could clarify or improve.
>
> Best,
>
> Authors of Paper 3711

---

> ### Comment · Reviewer_EKLC · 2022-12-06
> **Comments After Rebuttal**
>
> Thank you for fixing some items. However, the presentation of the material is still not clear. The primary confusing point is around equation (4) and the label allocation section. To move away from a trivial solution, one must rely on some statistics on ground truth labels or, use some assumptions. The paper still misses a discussion on such assumptions.

---

> > ### Author Response · Authors · 2022-12-06
> > **No Assumptions and Kindly Invite the Question Clarification**
> >
> >
> > Dear Reviewer EKLC,
> >
> > We appreciate your feedback that the partial concerns have been addressed in our first-round response, and sincerely thanks for your effort and time. However, we are confused about the case that the reviewer require us to place and discuss the assumptions. Currently, **our method has no assumptions on data and has never depended on the groundtruth labels but on the surrogate label statistics**. In [Table 1](https://openreview.net/forum?id=mqLowjofGBm&noteId=RFg5idCKh9) and [Table 3](https://openreview.net/forum?id=mqLowjofGBm&noteId=X6MkmLxBmA), we show how our method shows advantages on imbalanced data and degenerates to the ordinary contrastive learning in performance on balanced data.
> >
> > Overall, we do not require any assumptions when building our method, and can you **explain a bit that what assumptions you think that we may rely on**? Thank you very much again.
> >
> > Best
> >
> > The authors of Paper 3711

---

> > > ### Comment · Reviewer_EKLC · 2022-12-06
> > > **A though experiment**
> > >
> > > A simple though experiment shows that you have to have rely on some assumptions. Let's consider one dataset with two different Oracles and two different unbalanced labels. For example, we have a set of images with cats and dogs with the cat class being the most frequent. One Oracle knows which image has a cat or a dog. Also, assume same images are classified into 2 other classes: animals with floppy or pointy ears. Another Oracle know exact assignment of images based on the ear type. Assume that classes by both Oracles are not balanced. Now, let's apply your technique to the task of classifying images using cat/dog classes. Assume that it will find good surrogate labels and corresponding representation. Now, given that you did not use neither true labels, nor any assumptions, generated surrogate labels and corresponding representations should suit well the classification task by the second Oracle as well.
> > >
> > > I am not convinced that this conclusion holds true, thus the premise is incorrect which means you have to rely on some assumptions. You support your point using Table 1 and Table 3 tested on a specific task. You however claim a general applicability of your technique. Based on the though experiment above, I am not convinced that your results will hold on any unbalanced task. Understanding the limits of applicability (and thus assumptions) is crucial.

---

> > > > ### Author Response · Authors · 2022-12-07
> > > > **Response to The Example from Reviewer**
> > > >
> > > > We sincerely thank the reviewer for the detailed explanation and for the effort and time. To be clear, we first confirm the following contents to guarantee that we have accurately understood your concern: (1) "a though experiment" -->  "a thorough experiment"; (2) The data mentioned in the example can be classified from two perspectives of taxonomy, namely, **cat/dog** and **floppy ear/pointy ear**, and there is the underlying sample overlap in terms of two different taxonomies and the similar imbalance. For example, both cat images and floppy-ear images are majority, and one cat image might be with floppy ear (e.g., Scottish Fold) or pointy ear (e.g., caracal). If we are correct in this understanding, we have following clarification for your concerns.
> > > >
> > > >
> > > > > **Claim 1**: Now, let's apply your technique to the task of classifying images using cat/dog classes. Assume that it will find good surrogate labels and corresponding representation.
> > > >
> > > > Surrogate labels in **self-supervised learning** are in a pretty low granularity, and cannot be understood as the absolute representative of ground-truth labels in **supervised learning**. This is why we usually see the superior generalization ability of self-supervised learning, because the learning is not directly intervened by **labels in downstream supervised learning tasks**. Here, the surrogate labels are just from the intrinsic clustering patterns that can be richer and include both the features of two taxonomies. We use surrogate labels to adjust the geometric distribution of representation for the downstream-agnostic classification.
> > > >
> > > >
> > > > > **Claim 2**: Now, given that you did not use neither true labels, nor any assumptions, generated surrogate labels and corresponding representations should suit well the classification task by the second Oracle as well.
> > > >
> > > > In the following, we would like to kindly clarify:
> > > > - We **do not rely on surrogate labels to learn specific classification tasks**, but to rectify the representation geometry to encourage the linear identifiability, which improves linear probing.
> > > > - There are **no conflicts** between the two tasks.
> > > > - Critically, if this case is a problem, it is similarly applied to the balanced self-supervised learning.
> > > >
> > > > Overall, we would like to kindly clarify that **we have not involved more assumptions on data compared with the vanilla self-supervised learning**. We would like to discuss further with the reviewer. Thank you very much.
> > > >
> > > > Best
> > > >
> > > > The authors of 3711

---

> > > > > ### Comment · Reviewer_EKLC · 2022-12-08
> > > > > **Response**
> > > > >
> > > > > Thank you for the thorough response. To clarify, by " let's apply your technique to the task of classifying images using cat/dog classes" I meant using linear probing similar to what you reported in the tables.
> > > > >
> > > > >  > we have not involved more assumptions on data compared with the vanilla self-supervised learning
> > > > >
> > > > > And that is my point. Given that labels don't have to follow the geometric patterns that the method tries to capture, there are possible label distribution such that the proposed method will not perform well. In other words, there is no discussion on limitations of the approach.
> > > > >
> > > > > I am going to give a conditional acceptance, if you provide section on limitation of the approach. No method is universally good. Such a section will bring forth the assumption under which the method works well. By violating those assumption one can arrive at difficult examples.

---

> > > > > > ### Author Response · Authors · 2022-12-08
> > > > > > **Thanks for the response. We promise to add one section to discuss the limitation in the final revision.**
> > > > > >
> > > > > > Dear Reviewer EKLC,
> > > > > >
> > > > > > We sincerely thank the reviewer for the devoting time and efforts in reviewing the submission. We are so sorry that we misunderstand your intention in the early interaction, and promise to add one section that specially analyzes the limitations of $S^2LA$ and point out some potential direction that might be possible to further improve the method. Roughly, our design is built upon the intrinsic clustering patterns that can inclusively represent the information for the downstream tasks. Although we demonstrate the appealing performance in the current benchmark, it cannot be always guaranteed in all scenarios. Once such a condition is not satisfied, namely, clustering only captures the task-irrelevant patterns but ignores the task-relevant details, the improvement might be limited or even negative. A potential way to overcome this drawback is using a small auxiliary labeling set to calibrate the clustering dynamic aligned with the downstream tasks, namely, a semi-supervised paradigm. The methods to encourage learning the stable features in the area of causal inference can also be borrowed to this problem to alleviate this dilemma.
> > > > > >
> > > > > > Overall, we will detailedly discuss the potential limitations of our method and the corresponding possible solutions in the final version. Thank you very much!
> > > > > >
> > > > > > Best,
> > > > > >
> > > > > > Authors of Paper 3711

---

> > ### Author Response · Authors · 2022-12-07
> > **Would you mind clarifying your concerns? welcome for more discussions.**
> >
> > Dear Reviewer EKLC,
> >
> > Thanks again for the response. We have carefully considered your comments to improve our submission. In summary, we have (1) added more discussions on optimal transport with empirical evidence, (2) enriched the discussion and assumption to clarify where our method works and added supportive results, (3) clarified the intuition for logit adjustment and (4) carefully refined the mathematical definition. Besides, we also have provided our code anonymously. Would you mind **clarifying the unaddressed concerns w.r.t Q1-Q4** for further discussions and improvements of the manuscript. Thank you very much!
> >
> > Best,
> >
> > Authors of Paper 3711

---

### Official Review · Reviewer_1EQS · 2022-10-26

**Confidence:** 3
**Correctness:** 3
**Technical Novelty And Significance:** 2
**Empirical Novelty And Significance:** 3
**Recommendation:** 5

**Clarity, Quality, Novelty And Reproducibility:**

**Clarity and quality.** The description and motivation of the method is unclear, see concerns above. The empirical evaluation seems to be of a high standard. However, it would be good to perform multiple trials and include confidence intervals for the most important results.

**Novelty.** The method seems novel. Note that the use of Sinkhorn-Knopp for self-supervised learning via self-labelling was introduced in Asano et al. (2020). While this was not claimed as a contribution, I feel that this should be acknowledged in the paper.

**Reproducibility.** I believe there is sufficient information to reproduce the experiments.

**Strength And Weaknesses:**

**Strengths**

1. Dealing with long-tail data in unsupervised pre-training seems like an important and difficult problem which has not received much attention. Indeed, it's hard to even measure the tail since the dataset does not have labels.
1. Including focal loss as a baseline was good as it may help with class imbalance.
1. The proposed method is shown to significantly complement a wide range of algorithms for a wide range of datasets.
1. The method does not incur a high computational cost.

**Weaknesses**

1. The motivation and description of the Simplex ETF classifier, the transformation of its predictions to obtain surrogate labels and the additional loss term is extremely unclear. Similarly, the motivation for applying logit adjustment in reverse is unclear. If I understood correctly, the idea is to make the predictions of head classes even more confident, such that they are moved further from their (fixed) decision boundary. It might be possible to achieve this more elegantly using the logit-adjusted softmax from that same paper (rather than post-hoc logit adjustment)? Also, could the single $q \log p$ term in the loss be replaced with the more familiar cross-entropy over all labels $t$?
1. It wasn't clear what the "trivial unconstrained probabilities" and "trivial solution" to eq. 4 were. One-hot distributions? This was used to motivate the use of Optimal Transport. Can't we just use the (normalized) logit-adjusted likelihoods $p(\tilde{t})$ as targets? Does this cause the method to fail?
1. I found it surprising that the proposed approach achieved significant gains for the "Many" subset as well as the "Few" subset. This was not explained. Perhaps the method is achieving its improvement by some mechanism other than accounting for imbalance? Perhaps it's accounting for imbalance in some unlabelled attribute? This might be revealed by inspecting the clusters. Is it possible to run some experiments in the balanced setting and see if the method still yields gains?
1. No empirical comparison to the method of Asano et al. (SeLa), which does consider the imbalanced setting.
1. The values for $\tau$ seemed much lower than in the logit adjustment paper (0.02-0.07 vs. 1-4). How can this be explained?
1. While the method is claimed to be end-to-end, it involves non-differentiable computations in each epoch and in each mini-batch.
1. Does 300 iterations of Sinkhorn-Knopp really not have a significant impact on the training time? How long does this take compared to the typical time for a minibatch?
1. The number of training epochs seemed very high (e.g. for CIFAR-100-LT, 1000 epochs plus 500 warmup epochs). Is this typical for self-supervised training?
1. Figure 4(a), which shows a comparison to temperature scaling, would benefit from error-bars, since the values seem noisy and S2LA is not that far from the baseline.
1. Related work section does not include discussion of Supervised Long-tail Learning. For example, the paper "Decoupling Representation and Classifier for Long-Tailed Recognition" seems relevant.
1. Why was 100-shot evaluation used for ImageNet-LT and Places-LT? Wouldn't this impair the accuracy of the "Many" classes compared to a linear classifier trained with all examples?
1. While Algorithm 1 in the appendix makes the method much more clear, it shouldn't be necessary to refer to the appendix to understand the method. For example, it wasn't clear in the main text that the surrogate class prior was re-estimated once per epoch.

**Nitpicks**

1. Self-supervised learning includes some methods that have labels (i.e. automatically generated labels). The paper often uses "self-supervised learning" to refer to *contrastive* methods exclusively. It would be good to be more clear.
1. The acronyms BCL and SDCLR were not defined.
1. Needs to be proofread for grammar.
1. I was confused by the use of bold characters for the single labels $\boldsymbol{t}$ and $\tilde{\boldsymbol{t}}$. This is typically used to denote a vector.

**Summary Of The Paper:**

This paper considers the problem of self-supervised feature learning for downstream tasks where the unsupervised dataset follows a long-tail distribution.
In particular, it seeks to address the issue of head classes covering much more of the feature space than the tail classes (expansion/collapse).
The proposed method is to obtain initial surrogate class likelihoods for each example, transform these by applying reverse post-hoc logit adjustment based on the surrogate class frequencies, and then obtain final (soft) surrogate labels by solving an optimal transport problem with the Sinkhorn-Knopp algorithm.
The initial surrogate class likelihoods are obtained using a Simplex ETF (linear, constant, orthogonal) classifier and the final confidence of the assigned surrogate label is used to weight an additional loss term that encourages the Simplex ETF to be more confident.
The marginal likelihoods of the surrogate labels are recomputed every epoch, and the logit adjustment and optimal transport are computed in every minibatch.
Empirical studies show that this technique generally improves the accuracy of several baseline algorithms (SimCLR, Focal, SDCLR, BCL) across the Many, Med and Few class subsets for several datasets with a wide range of imbalance (CIFAR100-LT-10, -50, -100, ImageNet-LT, Places-LT).
It is applied after an initial warmup phase that uses the baseline algorithm alone.

**Summary Of The Review:**

While the results are strong, I find that the description and motivation of the method to be very unclear. I am leaning negative but I'll consider upgrading my review if these concerns can be mitigated in the process of the rebuttal and discussions.

---

> ### Author Response · Authors · 2022-11-16
> **Response to Reviewer 1EQS[3/3]**
>
> > **Q6**: Claims about the end-to-end manner.
>
> **A6**: To avoid ambiguity, we revise the draft with removing "end-to-end".
>
> > **Q7**: Computational cost.
>
> **A7**: Thanks for the reviewer’s question, we follow the suggestion to compare the computational cost and in the following table, we provide the mini-batch training time of different methods on CIFAR-100-LT, ImageNet-LT and Places-LT. In our runs, the proposed $S^2LA$ incurs an average $7.8\%$, $4.2\%$, $3.4\%$ computational overhead on CIFAR-100-LT, ImageNet-LT and Places-LT, respectively, which is relatively lightweight compared to the computational cost of deep neural networks. We have appended these results into the revision to help readers understand the computational cost of $S^2LA$ .
>
> [**Table 5.** The time cost of mini-batch training (seconds) on CIFAR-100-LT, ImageNet-LT and Places-LT.]
> | Dataset      | SimCLR | $+S^2LA$  | Focal | $+S^2LA$  | SDCLR | $+S^2LA$  |  BCL  | $+S^2LA$  |
> |--------------|:------:|:--------:|:-----:|:--------:|:-----:|:--------:|:-----:|:--------:|
> | CIFAR-100-LT |  0.379 |   0.407  | 0.423 |   0.467  | 0.374 |   0.397  | 0.377 |   0.405  |
> | ImageNet-LT  |  0.757 |   0.787  | 0.943 |   1.008  | 0.752 |   0.771  | 0.756 |   0.783  |
> | Places-LT    |  0.724 |   0.747  | 1.001 |   1.050  | 0.759 |   0.776  | 0.722 |   0.745  |
>
>
> > **Q8**: Training epochs.
>
> **A8**: Thanks for the reviewer’s question. For fair comparison with the baseline methods, we warmup 500 epochs and train $S^2LA$ 500 epochs, which is consistent with 1000 epochs for baseline methods on CIFAR-100-LT. Since the dataset size of the sub-sampled CIFAR-100 is a small, previous methods (SDCLR [2] and BCL [3]) adopt larger pre-training epochs (2000 epochs). In our experiment, we have decreased training epochs to 1000 but with the comparable performance.
>
>
> > **Q9**: Error bar for Figure 3(a).
>
> **A9**: Thanks for the suggestions, and following the reviewer's suggestion, we have added error bars to Figure 3(a) in the revision.
>
> > **Q10**:  Related works about supervised long-tailed learning.
>
> **A10**: We very appreciate the constructive advice of the reviewer and add more discussions about supervised long-tailed learning in the revision in the introduction and appendix B.
>
> > **Q11**:  100-shot evaluation on Large-scale dataset.
>
> **A11**: The 100-shot evaluation follows the setting in previous works (SDCLR [2] and BCL [3]). As shown in the following table, full-shot evaluation requires 10x - 30x the amount of data compared with the pre-training dataset, which might not be very practical. To further address the reviewer's concern, we provide 100-shot evaluation and full-shot evaluation on ImageNet-LT, as shown in the following table. We observe that the performance improvements and representation balancedness (Std) are consistent with the 2 evaluations, indicating the rationality of the 100-shot evaluation.
>
>
> [**Table 6.** Statistics of long-tailed datasets.]
> | Dataset     | \# Class | \# Training data | \# 100-shot data | \# full-shot data | \# Test data |
> |-------------|:--------:|:----------------:|:----------------:|:-----------------:|:------------:|
> | ImageNet-LT |   1000   |      115,846     |      100,000     |     1,261,167     |    50,000    |
> | Places-LT   |    365   |      62,500      |      36,500      |     1,803,460     |    36,500    |
>
> [**Table 7.** Evaluation on ImageNet-LT.]
> | Evaluation | Method   |  Many | Medium |  Few  |  Std |  Avg  |
> |------------|----------|:-----:|:------:|:-----:|:----:|:-----:|
> | 100-shot   | SimCLR   | 41.69 |  33.96 | 31.82 | 5.19 | 36.65 |
> |            | $+S^2LA$ | 41.53 |  36.35 | 35.84 | 3.15 | 38.28 |
> | Full-shot  | SimCLR   | 42.86 |  35.17 | 33.13 | 5.13 | 37.86 |
> |           | $+S^2LA$ | 44.11 |  38.59 | 37.87 | 3.41 | 40.62 |
>
>
> > **Q12**:  More details about the algorithms in the main body.
>
> **A12**: We very appreciate the advice of the reviewer and revise the draft for more details in Section 4, after the "Label Allocation" part.
>
> > **Q13**: Minor issue.
>
> **A13**: Thank you for all detailed comments on the writting. We revise each of them according to the reviewer's advice and carefully proofread the whole paper again. Please refer to the corresponding revision in the draft. Besides, BCL and SDCLR is defined in Section 5.1. As for the labels $\boldsymbol{t}$ and $\tilde{\boldsymbol{t}}$, we use bold characters for clearer understanding of the marginals $p(\boldsymbol{t})$ on the population level, where $\boldsymbol{t}$ denotes the prediction vector on the dataset.
>
> [1] Asano Y M, Rupprecht C, Vedaldi A. Self-labelling via simultaneous clustering and representation learning. International Conference on Learning Representations. 2019.
>
> [2] Jiang Z, Chen T, Mortazavi B J, et al. Self-damaging contrastive learning. International Conference on Machine Learning. 2021.
>
> [3] Zhou Z, Yao J, Wang Y F, et al. Contrastive Learning with Boosted Memorization. International Conference on Machine Learning. 2022.

---

> ### Author Response · Authors · 2022-11-16
> **Response to Reviewer 1EQS[2/3]**
>
> [**Table 2.** Normalized mutual information (NMI) between the geometric predictions and the oracle labels]
> | Prediction | $\tilde{\boldsymbol{t}}$ with Post-hoc (CE) | $q(\boldsymbol{t} \vert \boldsymbol{x})$ with $S^2LA$  |
> |------------|:--------------------------------:|:--------------------------:|
> | NMI score  |               0.056              |            0.432           |
>
> As can be seen, both the logit-adjusted losses and the post-hoc logit-adjusted manners (I1-I4) lead to performance degradation compared with the vanilla SimCLR, while the label allocation by optimal transport ($S^2LA$) significantly improves the performance. To explain this, we compute the NMI score between $\tilde{\boldsymbol{t}}$ and the oracle label, and the NMI score between $q(\boldsymbol{t}|\boldsymbol{x})$ and the oracle label, summarized in the below table. Accorrding to the comparison of their NMIs, we can understand the roughly adjusted $\tilde{\boldsymbol{t}}$ can be noisy and shows the weak correlation with the oracle label, while $q(\boldsymbol{t}|\boldsymbol{x})$ reallocated by optimal transport based on the population statistic of $\tilde{\boldsymbol{t}}$ effectively improves the correlation.
>
>
> We have added these results and discussions about the merit of optimal transport for label refinement in the submission to make this more clear.
>
> [Explainations about ''unconstrained probabilities'' and ''trivial solution''.]
>
> Thank you for the detailed comments. The ''unconstrained probabilities'' means that the predictions obtained by the straightforward instance-level adjustment can be noisy. The ''trivial solution'' indicates the severe performance degradation of the model. From the above results, we can see that the joint optimization setup only achieves 2.91\% accuracy in the linear probing evaluation.
>
> > **Q3**: Explanations of performance gain on disjoint groups.
>
> **A3**: We would like to kindly clarify that our method achieves more significant gains on the tail group, e.g.  1.41\%/2.32\%/3.24\% on CIFAR-100-LT-R100 and 0.43\%/2.30\%/3.50\% on ImageNet-LT.
> However, we appreciate the reviewer's concern and follow the reviewer's advice to conduct experiments in the balanced setting, as shown in the following table. From the results, we can see that $S^2LA$ shows comparable performance with the baseline methods in the balanced setting. This indicates that the effectiveness of $S^2LA$ mainly comes from accounting for the data imbalance.
>
>
> [**Table 3.** Ablation study in the balanced setting on CIFAR-100-LT-R100.]
> | Method   | SimCLR | $+S^2LA$  | Focal | $+S^2LA$  | SDCLR | $+S^2LA$  |  BCL  | $+S^2LA$  |
> |----------|:------:|:--------:|:-----:|:--------:|:-----:|:--------:|:-----:|:--------:|
> | Accuracy |  66.75 |   66.41  | 66.42 |   66.79  | 65.96 |   66.17  | 69.16 |   69.33  |
>
>
> > **Q4**: Comparisons with SeLa.
>
> **A4**: Thanks for recommending this wonderful work. We have included SeLa [1] as a baseline method into the revised draft with the corresponding discussion (Section 2). In the following, we use the official code of SeLa with the default training settings (marked as SeLa), with our training schedule (marked as SeLa*) and conduct the experiments on CIFAR-100-LT-R100. From the results, we can see that our method can improve the performance of SeLa.
>
> [**Table 4.** Empirical comparisons with SeLa on CIFAR-100-LT-R100.]
> | Method   |  SeLa | SeLa* | SeLa*  $+S^2LA$  |
> |----------|:-----:|:-----:|:---------------:|
> | Accuracy | 44.45 | 46.47 |      48.10      |
>
> > **Q5**: Scale of hyperparameters $\tau$.
>
> **A5**: The explanation about such a difference is that LA operates on the logit whose scale is unbounded, while our method operates on cosine similarity whose scale is between [-1,1].

---

> ### Author Response · Authors · 2022-11-16
> **Response to Reviewer 1EQS[1/3]**
>
> Thank you for your time devoted to reviewing this paper and your constructive suggestions. Here are our detailed replies to your questions.
>
> > **Q1 (1)**: Motivation and description of Simplex ETF classifier.
>
> **A1 (1)**: Our motivation is to make self-supervised learning robust to the class-imbalance data, which requires the pursuit in the embedding space intrinsically switching from the sample-level uniformity to the category-level uniformity. The Simplex ETF is a tool to measure the gap between the category-level uniformity and the sample-level uniformity, which is then transformed as the supervision feedback to the training. We have comprehensively modified the submission to make them more clear.
>
> > **Q1 (2)**: Motivation of logit adjustment.
>
> **A1 (2)**: As the reviewer's understanding, logit adjustment here is to strengthen the estimation of samples from head classes so that they can be discovered as many as possible. Then, the feature regime of head-class samples can be sufficiently suppressed via Eq.(7) to pursue the category-level uniformity. This is quite different from logit adjustment in supervised long-tailed learning, which is to calibrate the quantity bias towards different classes of samples in the logit space. We have added more explanations near Eq.(3) to clarify this point.
>
>
> > **Q1 (3) + Q2**: Ablation on Optimal Transport.
>
> **A2**: We appreciate the advice of the reviewer. In the following, we conduct a range of experiments to verify the effectiveness of our design, which includes (I-1) the logit-adjusted loss with soft assignments, (I-2) the logit-adjusted loss with hard assignments, (I-3) post-hoc logit-adjusted loss with soft assignments, (I-4) post-hoc logit-adjusted loss with hard assignments. Besides, we also add two methods: joint optimization on $p$, $q$ and oracle ground-truth guidance.
>
> [**Table 1.** Ablation study on optimal transport]
> | Method                         | Acronym |                                                                              Formulation                                                                             | Accuracy |
> |--------------------------------|:-------:|:--------------------------------------------------------------------------------------------------------------------------------------------------------------------:|:--------:|
> | SimCLR                         |    -    |                                                                                   -                                                                                  |   50.72  |
> | +Joint Optimization            |    -    |              $\min_{p,q} -\frac{1}{\vert\mathcal{D}\vert} \sum_{x \sim \mathcal{D}}  q(\boldsymbol{t}\vert\boldsymbol{x}) \log p(\boldsymbol{t}\vert\boldsymbol{x})$             |   2.91   |
> | +Logit-adjusted softmax (soft) |   I-1   | $\min_{p} -\frac{1}{\vert\mathcal{D}\vert} \sum_{x \sim \mathcal{D}}  p(\boldsymbol{t}\vert\boldsymbol{x}) \log p(\boldsymbol{t}\vert\boldsymbol{x}) / p(\boldsymbol{t})^{\tau}$ |   48.41  |
> | +Logit-adjusted softmax (hard) |   I-2   |           $\min_{p} -\frac{1}{\vert\mathcal{D}\vert} \sum_{x \sim \mathcal{D}}  \boldsymbol{t} \log p(\boldsymbol{t}\vert\boldsymbol{x})/ p(\boldsymbol{t})^{\tau}$           |   49.93  |
> | +Post-hoc (soft)               |   I-3   |           $\min_{p} -\frac{1}{\vert\mathcal{D}\vert} \sum_{x \sim \mathcal{D}}  p(\tilde{\boldsymbol{t}}\vert\boldsymbol{x}) \log p(\boldsymbol{t}\vert\boldsymbol{x})$          |   50.38  |
> | +Post-hoc (hard)               |   I-4   |                    $\min_{p} -\frac{1}{\vert\mathcal{D}\vert} \sum_{x \sim \mathcal{D}}  \tilde{\boldsymbol{t}} \log p(\boldsymbol{t}\vert\boldsymbol{x})$                    |   50.22  |
> | +$S^2LA$                       |    -    |              $\min_{p,q} -\frac{1}{\vert\mathcal{D}\vert} \sum_{x \sim \mathcal{D}}  q(\boldsymbol{t}\vert\boldsymbol{x}) \log p(\boldsymbol{t}\vert\boldsymbol{x})$             |   53.96  |
> | +Oracle                        |    -    |                      $\min_{p} -\frac{1}{\vert\mathcal{D}\vert} \sum_{(x,y) \sim \mathcal{D}}  \boldsymbol{y} \log p(\boldsymbol{t}\vert\boldsymbol{x})$                      |   56.47  |

---

> ### Author Response · Authors · 2022-11-17
> **Looking forward to your response**
>
> Dear Reviewer 1EQS,
>
> We sincerely thank you for your great efforts in the review of our manuscript. As the discussion period is approaching its end, we would be grateful if you could confirm whether our responses and the additions we have made to the draft addressed your concerns, and let us know if any issues remain.
>
> Best,
>
> Authors of Paper 3711

---

> ### Author Response · Authors · 2022-11-18
> **Would you mind checking our response? welcome for more discussions.**
>
> Dear Reviewer 1EQS,
>
> We sincerely thank you for your great efforts in the review of our manuscript! As the Stage 1 Discussion period is approaching its end, here is a summary of our previous response and update:
>
> - Revised the motivation of Simplex ETF (see Section 3.3, Section 4) and logit adjustment (see Section 4).
> - Expanded the discussion about optimal transport (see Section 4) with more empirical evidence (see Section 5.4).
> - Expanded related works about supervised longh-tailed learning (see Appendix B).
> - Added discussion (see Section 2) and comparisons (see Appendix E.4) with SeLa.
> - Added the results about the computational cost (see Section 5) and error bar for Figure 3(a).
> - Expanded the experimental evaluation details (see Appendix D.2) with proper empirical results on large-scale datasets (see Appendix E.5).
> - Clarified the performance gain, training epochs and added empirical results in the balanced setting (see Appendix C).
>
> We would be grateful  if you could check our responses with the updated manuscript, and confirm whether our responses have addressed your concerns. Please let us know if there are any further questions or suggestions that we could clarify or improve. More discussions are always welcome!
>
> Best,
>
> Authors of Paper 3711

---

> > ### Comment · Reviewer_1EQS · 2022-11-21
> > **Much better but communication of ideas still needs to be improved**
> >
> > Thank you for taking the time to refine the manuscript, highlight the changes, and present further exploratory results.
> >
> > I'm satisfied with: the speed of the optimal transport computation, the comparison to post-hoc logit adjustment (no optimal transport) in Table 10, related work about supervised long-tail learning, the comparison to SeLa and the errorbars.
> >
> > While it's good that Table 10 is cited as motivation for introducing optimal transport, it would help to also provide some explanation of the reason that this occurs; i.e. what problem does the use of optimal transport solve compare to simple post-hoc logit adjustment?
> > I also find it puzzling that "post-hoc logit adjustment" is similar to "S2LA" in Table 10 but much worse in Table 11 (NMI).
> >
> > It feels like logit adjustment is being used to create a kind of positive feedback loop that increases the concentration of frequent (pseudo-) classes in the feature space.
> > Intuitively, this seems somewhat risky as the feedback loop could be unstable?
> >
> > It's not clear whether features are L2-normalised (projected to the hypersphere) such that inner products represent geodesic (cosine) distances.
> > I believe that they are?
> > If so, stating this explicitly would improve the clarity of the paper.
> >
> > There are frequent passages in the paper that are difficult to understand.
> > The authors often employ terminology that may be clear to them, but is not clear to a wider audience.
> > Some examples include "feature regime", "strengthen the estimation of samples", "use Simplex ETF to measure the embedding space".
> > As a result, the motivation of each step is not always clear.
> >
> > Overall, this paper seems to identify a real issue and propose a working solution.
> > However, the exposition of the problem and the solution need to be improved for it to be useful to the community.
> > I believe that improving the writing will help to frame the solution in a more elegant fashion.
> > I still lean towards reject, although I would not be upset if the paper were accepted.

---

> > > ### Author Response · Authors · 2022-11-21
> > > **Thanks for the response**
> > >
> > > Dear Reviewer 1EQS,
> > >
> > > Thank you very much for your constructive comments with your devoted time and efforts. Here are our detailed replies to your further questions.
> > >
> > > - As shown in Table 11, we observe that the generated predictions with post-hoc LA shows weak correlations with the ground-truth labels, which means that the straightforward instance-level adjusted predictions are still very noisy. This motivates us to use optimal transport based on the prior accumulated by the post-hoc LA predictions to further refine the label posteriors. Similarly according to the NMI in Table 11, the re-allocated predictions are more correlated with the oracle labels, thus showing better performance. As for the results in Table 10, we would like to kindly clarify that the proposed method (53.96\%) also significantly outperforms post-hoc LA (50.22\%) in terms of accuracy.
> > > - We would like to kindly clarify that our method is working jointly with the contrastive learning baselines. Specially, the proposed method encourages the space shrinking of head classes while the contrastive baselines sort to distribute the data points uniformly. This fullfills a positive collaboration to achieve the desired representation.
> > > - We will add more explanations and descriptions about the terminology to ease the understanding. We will also further proofread the manuscript in the final revision. Besides, we will follow reviewer's advice to state that the features are L2-normalised.
> > >
> > > Best,
> > >
> > > Authors of Paper 3711

---

### Author Response · Authors · 2022-11-16
**General Response by Authors**

We would like to thank all the reviewers for their constructive suggestions on our submission. We are glad that the reviewers have some positive impressions of our work, including: (1) an interesting and important research problem (1EQS, EKLC, jKEp), (2) novel and original idea (1EQS, EKLC, bvPJ) (3) clear motivation and well-defined mathematical principles (jKEp, bvPJ), (4) strong and comprehensive experiments (1EQS, jKEp, bvPJ).

According to the advice, we have carefully revised our draft with the proofreading to correct some typos and mistakes, and complete a range of experiments to address the concerns of the reviewers. In the following, we provide a summary of our updates, and for detailed responses, please refer to the feedback of each comment/question and the new empirical evaluations.

1. We carefully refine the motivation and description of Simplex ETF and logit adjustment to avoid the potential misunderstanding on the novelty (See Section 4), and specially add more explainations to back up the rationality of the optimal transport (See Section 4) with a variants of empirical comparisons for further understanding (See Section 5.4).

2.  We enrich the description in the preliminary parts (See Section 3) and refine the mathematical definition to ease the understanding (See Section 3.1, Section 4), add necessary notations and more discussions for Eq. (4) with the empirical support (See Section 5.4), about its rationality and intuition, and complete the implementation details of our algorithm in the main body (See Section 4).

3. We conduct more ablation studies to verify the effectiveness of our method, including: (1) Ablation study on optimal transport (See Section 5.4), (2) Balanced data (See Appendix C), (3) $S^2LA$ random classifier (See Section 5.4), (4) Simplex ETF alone as an projector (See Section 5.4), (5) Comparisons on computational cost (See Section 5.4), (6) Full-shot evaluation on the large-scale dataset (See Appendix E.5).

The above updates in the revised draft (the regular pages and the appendix) are highlighted in blue color.

We appreciate all reviewers’ great effort again. We are looking forward to your reply!

---

### Decision · Program_Chairs · 2023-01-20

**Decision:**

Reject

**Justification For Why Not Higher Score:**

Lack of clarity of method. Proper discussion and analysis of limitations and assumptions.

**Justification For Why Not Lower Score:**

N/A

**Metareview: Summary, Strengths And Weaknesses:**

This paper tackles the problem of unbalanced classes in self-supervised learning. Authors posits that imbalanced classes (long-tailed data) can lead to the model producing higher quality embedding for the head samples (of high frequency classes) and lower quality embedding for tail samples (of low frequency classes). This problem has been studied recently in prior work. Authors propose a new training strategy which aims to balance out (equally) the spread of various underlying classes on the embedding space. It does this by creating surrogate labels for the unlabeled data, measuring the imbalance and then correcting it through an adjustment of the surrogate logits.

The method uses some interesting tools and is quite flexible that it can work on top (in addition to) of prior method. Authors provide experiments which shows that their method improves the downstream-task performance on top of other baseline method. Reviewer requested for additional experiments and clarification and authors tried to address many of these concerns. Unfortunately, it is notable that none of numerical results in the table provide mean and/or standard errors over multiple runs, so there is still some uncertainty about the level of improvement. All reviewers had mentioned that the proposed method is hard to understand (Sec 4) due to jargon/imprecise wording, and lack of rigorous mathematical derivation. Currently it takes considerable effort to understand even the basic details of the method. This concern still remained even after the revisions. Finally, a reviewer also noted no limitations of the method was discussed which was concerning. This lead to a discussion between authors and reviewer which brought up an interesting failure case when there could be multiple underlying class taxonomies, and authors have promised add them to the main text.